# ONE OBJECTIVE FOR ALL MODELS – SELF-SUPERVISED LEARNING FOR TOPIC MODELS

## ABSTRACT

Self-supervised learning has significantly improved the performance of many NLP tasks. In this paper, we highlight a key advantage of self-supervised learning - when applied to data generated by topic models, self-supervised learning can be oblivious to the specific model, and hence is less susceptible to model mis-specification. In particular, we prove that commonly used self-supervised objectives based on reconstruction or contrastive samples can both recover useful posterior information for general topic models. Empirically, we show that the same objectives can perform competitively against posterior inference using the correct model, while outperforming posterior inference using mis-specified model.

## 1 INTRODUCTION

Recently researchers have successfully trained large-scale models like BERT (Devlin et al., 2018) and GPT (Radford et al., 2018), which offers extremely powerful representations for many NLP tasks (see e.g. Liu et al. (2021); Jaiswal et al. (2021) and references therein). To train these models, often one starts with sentences in a large text corpus, mark random words as "unknown" and ask the neural network to try to predict the unknown words. This approach is known as self-supervised learning. Despite many attempts, there is still no complete understanding of why self-supervised learning would lead to useful representations. A major bottleneck is that a "useful" representation often has no precise mathematical definition in the practical settings.

One approach to get a precise definition for "useful" representation is to apply self-supervised learning on synthetic data generated by probabilistic models. For example, many topic models (such as Latent Dirichlet Allocation (LDA) (Blei et al., 2003), Correlated Topic Model (CTM) (Blei & Lafferty, 2007), Pachinko Allocation Model (PAM) (Li & McCallum, 2006)) treat documents as a bag-of-words and use a two-step procedure of generating a document: first each document is viewed as a probability distribution of topics (often called the topic proportions $w$), and each topic is viewed as a probability distribution over words. If we use a matrix $A$ to represent a topic-word matrix, where $A_{ij}$ represents the probability of using word $i$ in topic $j$, then the words in this document will have probabilities according to $Aw$. Since the topic-word matrix $A$ is shared across the entire corpus, and the topic distribution vector is specific to documents, it makes sense to define $w$ as the representation for the document. In this setting, the questions we try to answer are: *Can self-supervised learning learn a representation that contains information about $w$? Why is self-supervised learning better than the traditional approach of learning the probabilistic models and applying inference?*

In this paper, we show that if the standard reconstruction-based objective (see Equation (1), similar to the objective used in Pathak et al. (2016); Devlin et al. (2018)) can be minimized, then the network will necessarily learn information about posterior of $w$ (see Theorem 3). We also highlight one particular benefit of self-supervised learning – as long as the objective can be minimized, self-supervised learning does not depend on what topic model was used to generate the data, while traditional inference methods rely heavily on having the correct model. Self-supervised learning is much more robust to model mis-specification.

Of course, the theory still needs to assume that either self-supervised learning objective can be successfully minimized (or at least approximately minimized). We demonstrate that this is indeed the case in practice in Section 5. In particular, we consider four different topic models (pure, LDA, CTM and PAM) and show that using the same reconstruction-based objective, it can extract information about the topic proportion $w$. Our experiments show that the performance of the self-supervised

model is competitive against posterior inference using the correct model, while outperforming posterior inference using mis-specified model. However, our experiments also show that it can be beneficial to adapt the architecture of the network to the complexity of the topic model (i.e., larger/more complex networks for CTM/PAM compared to pure and LDA models). Therefore if we take optimization into consideration, self-supervised learning is still more robust but not completely oblivious to the exact topic model. Finally in Section 6, we also demonstrate that minimizing this objective can lead to meaningful representations for real datasets (which are not generated by topic models).

## 1.1 RELATED WORKS

**Self-Supervised Learning** Self-supervised learning recently has been shown to be able to learn useful representation, which is later used for downstream tasks. See for example Bachman et al. (2019); Caron et al. (2020); Chen et al. (2020a;b;c); Grill et al. (2020); Chen & He (2021); Tian et al. (2020a); He et al. (2020) and references therein. In particular, Devlin et al. (2018) proposed BERT, which shows that self-supervised learning has the ability to train large-scale language models and could provide powerful representations for downstream natural language processing tasks.

**Theoretical Understanding of Self-Supervised Learning** Given the recent success of self-supervise learning, many works have been tried to provide theoretical understanding on contrastive learning (Arora et al., 2019; Wang & Isola, 2020; Tosh et al., 2020; Tian et al., 2020b; HaoChen et al., 2021; Wen & Li, 2021; Zimmermann et al., 2021) and reconstruction-based learning (Lee et al., 2020; Saunshi et al., 2020; Teng & Huang, 2021). Also, several papers considered the problem from a multi-view perspective (Tsai et al., 2020; Tosh et al., 2021), which covers both contrastive and reconstruction-based learning. Moreover, Wei et al. (2020) and Tian et al. (2021) studied the theoretical properties of self-training and the contrastive learning without the negative pairs respectively. Saunshi et al. (2020) investigated the benefits of pre-trained language models for downstream tasks. They showed that if one can predict the next word accurately, then the learned representation can be used to solve downstream linear task. Our setting is different from theirs as we focus on the probabilistic models. Most relevant to our paper, Tosh et al. (2020) considered the contrastive learning in the topic models setting. Our theoretical results extend their theory to reconstruction-based objective (while also removing some assumptions), and our empirical results show that the reconstruction-based objective can be effectively minimized.

**Theoretical Analysis of Topic Models** Topic models are popular probabilistic models that has proved successful in many applications while still have clear structure to allow theoretical studies. Many works have proposed provable algorithms to learn topic models, such as method of moment based approaches (Anandkumar et al., 2012; 2013; 2014; 2015) and anchor word based approaches (Papadimitriou et al., 2000; Arora et al., 2012; 2016a; Gillis & Vavasis, 2013; Bittorf et al., 2012). Much less is known about provable inference for topic models. Sontag & Roy (2011) showed that MAP estimation can be NP-hard even for LDA model. Arora et al. (2016b) considered approximate inference algorithms.

## 1.2 OUTLINE

We first introduce the basic concepts of topic models and our objectives in Section 2. Then in Section 3 we prove guarantees for the reconstruction-based objective. Section 4 connects the contrastive objective to reconstruction-based objective which allows us to prove a stronger guarantee for the former. We then demonstrate the ability of self-supervised learning to adapt to different models in synthetic experiments in Section 5. Finally, we also evaluate the reconstruction-based objective on real-data to show that it extracts high-quality representations.

## 2 PRELIMINARIES

In this section we first introduce some general notations. Then we briefly describe the topic models we consider in Section 2.1. Finally we define the self-supervised learning objectives in Section 2.2 and give our main results.

**Notation** We use $[n]$ to denote set $\{1, 2, \ldots, n\}$. For vector $x \in \mathbb{R}^d$, denote $\|x\|$ as its $\ell_2$ norm and $\|x\|_1$ as its $\ell_1$ norm. For matrix $A \in \mathbb{R}^{m \times n}$, we use $A_i \in \mathbb{R}^m$ to denote its $i$-th column. When

matrix $A$ has full column rank, denote its left pseudo-inverse as $A^\dagger = (A^\top A)^{-1} A^\top$. For matrix or general tensor $T \in \mathbb{R}^{d_1 \times \cdots \times d_l}$, we use vector $\text{vec}(T) \in \mathbb{R}^{d_1 \cdots d_l}$ to represent its vectorization. Let $\Delta^K$ to denote $K - 1$ dimensional probability simplex. For two probability vectors $p, q$, define their total variation (TV) distance as $\text{TV}(p, q) = \|p - q\|_1 / 2$.

## 2.1 TOPIC MODELS

As we mentioned earlier, the generative process of many topic models can be viewed as a two-step procedure. We consider a general scenario where $\mathcal{V}$ is a finite set of vocabulary with size $V$ and $\mathcal{K}$ is a set of $K$ topics, where each topic is a distribution over $\mathcal{V}$. We denote the topic-word matrix as $A$ with $A_{ij} = \mathbb{P}(\text{word } i \mid \text{topic } j)$, so that each column in $A$ represents the word distribution of a topic. Each document corresponds to a convex combination of different topics with proportions $w$. For each word in the document, first sample a topic $z$ according to the topic proportions $w$, and then sample the word from the corresponding topic (equivalently, one can also sample a word from distribution $Aw$). Different topic models differ in how they generate $w$, which we formulate in the following definition:

**Definition 1** (General Topic Model). *A general topic model specifies a distribution $\Delta(K)$ for each number of topics $K$. Given a topic matrix $A$ and $\Delta(K)$, to generate a document, one first sample $w \sim \Delta(K)$ and then sample each word in the document from the distribution $Aw$.*

Here $\Delta(K)$ is a prior distribution of $w$ which is crucial when we are trying to infer the true topic proportions $w$ given the words in a document. Of course, different topic models may also specify different priors for the topic matrix $A$. However, given a large number of documents generated from the topic model, in many settings one can hope to learn the topic matrix $A$ and the prior distribution $\Delta(K)$ (see e.g., Arora et al. (2012; 2016a)), therefore we consider the following inference problem:

**Definition 2** (Topic Inference). *Given a topic matrix $A$, prior distribution $\Delta(K)$ for topic proportions $w$, and a document $x$, the topic inference problem tries to compute the posterior distribution $w|x, A$ given the prior $w \sim \Delta(K)$.*

Note that our general topic model can capture many standard topic models, including pure topic model, LDA, CTM and PAM.

**Semi-supervised Learning Setup**  As our goal is to understand the representation learning aspect of self-supervised learning, we consider a semi-supervised setting where each document also has a label $y$ which is determined as a function of $w$. We are given large number of unlabeled documents and a small number of labeled documents, the goal is to apply self-supervised learning on the unlabeled documents to get a good representation, and then use this representation to predict the label $y$. Of course, if one knows the actual parameters of the model, the best way predictor for $y$ would be to first estimate the posterior distribution $w$ and then apply the function that maps $w$ to label $y$. We will show that for functions that are approximable by low degree polynomials self-supervised learning can always provide a good representation.

## 2.2 SELF-SUPERVISED LEARNING

In general, self-supervised learning tries to turn unlabeled data into a supervised learning problem by hiding some known information. There are many ways to achieve this goal (see e.g., Liu et al. (2021); Jaiswal et al. (2021)). In this paper, we focus on two different approaches for self-supervised learning: reconstruction-based objective and contrastive objective. We first formally define the objective and then discuss our corresponding result.

**Reconstruction-Based Objective**  One common approach of self-supervised learning is to first mask part of the data and then try to find a function $f$ to reconstruct the missing part given the unmasked part of input. This is commonly used for language modeling. In the context of topic modelling, since each word is i.i.d. sampled, given the document $x_{\text{unsup}}$ that generated from word distribution $Aw$, we pick $t$ random words from the documents and mark them as unknown, then we ask the learner to predict these $t$ words given the remaining words in the documents. Specifically, let $y$ be the $t$ words that we select and let $x$ be the document with these $t$ words removed, we aim to

select a predictor $f$ that minimizes the following reconstruction objective:

$$\min_f L_{\text{reconst}}(f) \triangleq \mathbb{E}_{x,y}[\ell(f(x), y)], \tag{1}$$

where $\ell(\hat{y}, y) = \sum_k -y_k \log \hat{y}_k$ is the cross entropy loss. Here we slightly abuse the notation to use $y$ as an one-hot vector for $|\mathcal{V}|^t$ classes and $f$ also outputs a probability distribution on $\mathcal{V}^t$. Depending on the context, we will use $y$ to denote either the actual next $t$ words or its corresponding one-hot label. Now we are ready to present our result for the reconstruction-based objective:

**Theorem 1.** *(Informal) Consider the general topic model setting as Definition 1, suppose function $f$ minimizes the reconstruction-based objective* (1). *Then, any polynomial $P(w)$ of the posterior $w|x, A$ with degree at most $t$ can be represented by a linear function of $f(x)$.*

The theorem shows that if we want to get basic information about posterior distribution (such as mean, variance), then it suffices to predict a small constant number of words (1 for mean and 2 for variance). See Theorem 3 for the formal statement.

**Contrastive Objective** Another common approach in self-supervised learning is the contrastive learning. In contrastive learning, the training data is usually a pair of data $(x, x')$ with a label $y \in \{0, 1\}$, where label 1 means $(x, x')$ is a positive sample ($x$ and $x'$ are similar) and label 0 means $(x, x')$ is a negative sample ($x$ and $x'$ are dissimilar). The task is to find a function $f$ such that it can distinguish the positive sample and negative sample, i.e., $f(x, x') = y$. Formally, we want to select a predictor $f$ such that it minimizes the following contrastive objective:

$$\min_f L_{\text{contrast}} \triangleq \mathbb{E}_{x,x',y}[\ell(f(x, x'), y)]. \tag{2}$$

In the context of topic models, following previous work (Tosh et al., 2020), we generate the data $(x, x', y)$ as follows. We first generate a document $x$ from the word distribution $Aw$, then (i) with half probability we generate $t$ words from the same distribution $Aw$ to form the document $x'$ and set $y = 1$; (ii) with half probability we generate $t$ words from a different word distribution $Aw'$ with $w' \sim \Delta(\mathcal{K})$ (so that $w \neq w'$) and set $y = 0$. For loss function $\ell$, we consider the square loss $\ell(\hat{y}, y) = (\hat{y} - y)^2$. However, our results hold for other loss function such as logistic loss.

We now give our informal result on contrastive objective. See Theorem 5 for the formal statement. Note that this theorem generalizes Theorem 3 in Tosh et al. (2020).

**Theorem 2.** *(Informal) Consider the general topic model setting as Definition 1, suppose function $f$ minimizes the contrastive objective* (2). *Then we can use $f$ and enough documents to construct a representation $g(x)$ such that any polynomial $P(w)$ of the posterior $w|x, A$ with degree at most $t$ can be represented by a linear function of $g(x)$.*

## 3 GUARANTEES FOR THE RECONSTRUCTION-BASED OBJECTIVE

In this section, we consider the reconstruction-based objective (1) and provide theoretical guarantees for its performance. We first show that if such objective with $t$ unknown words can be minimized, then any polynomial of topic posterior $w|x, A$ with degree at most $t$ can be represented by a linear function of the learned representation.

**Theorem 3** (Main Result). *Consider the general topic model setting as Definition 1, suppose topic matrix $A$ satisfies rank$(A) = K$ and function $f$ minimizes the reconstruction-based objective* (1). *Then, any polynomial $P(w)$ of the posterior $w|x, A$ with degree at most $t$ is linear in $f(x)$, that is there exists a $\theta \in \mathbb{R}^{V^t}$ such that for all documents $x$*

$$\mathbb{E}_w[P(w)|x] = \theta^\top f(x).$$

To understand this theorem, we can first think about a warm-up example where $t = 1$ (see Section A.1 in appendix). Intuitively, in this case the best way to predict the missing word is to estimate the topic proportion $w$, and then predict the word using $Aw$ where $A$ is the topic matrix. Therefore, if the output of self-supervised learning $f(x)$ (which is a $V$-dimensional vector indexed by words) minimizes the loss, then $f(x)$ must have the form $f(x) = A\mathbb{E}[w|x]$. When $A$ is full rank multiplying by the pseudo-inverse of $A$ recovers the expectation of the posterior. The proof for the general case of predicting $t$-words requires more careful characterization of the optimal prediction and its relationship to $\mathbb{E}_w[P(w)|x]$, which we defer to Section A.2.

**Robustness for an approximate minimizer**  In Theorem 3 we focus on the case when function $f$ is exactly the minimizer of reconstruction-based objective (1), i.e., $L_{\text{reconst}}(f) = L^*_{\text{reconst}} \triangleq \min_f L_{\text{reconst}}(f)$. However, in practice one cannot hope to find such a function exactly. In the following, we provide a robust version of Theorem 3 such that it allows us to find an approximate solution instead of the exact optimal solution.

To present our result, we need to first introduce the following notion of condition number, which was used in many previous works, such as collaborative filtering system (Kleinberg & Sandler, 2008) and topic models (Arora et al., 2016b). Intuitively, $\kappa(B)$ measures how large a vector would change after multiplying with $B$ in the $\ell_1$ norm sense.

**Definition 3** ($\ell_1$ Condition Number). *For matrix $B \in \mathbb{R}^{m \times n}$, define its $\ell_1$ condition number $\kappa(B)$ as $\kappa(B) \triangleq \min_{x \in \mathbb{R}^n : x \neq 0} \|Bx\|_1 / \|x\|_1 = \max_{i \in [n]} \|B_i\|_1$, where $B_i$ is the $i$-th column of $B$.*

Let $W_{post} \in \mathbb{R}^{K \times \cdots \times K}$ be the topic posterior tensor for the $t$ unknown words $y = (y_1, \ldots, y_t)$ given document $x$. That is, for each entry $[W_{post}]_{z_1, \ldots, z_k} = \mathbb{P}(z_i$ is the topic of word $y_i$ for $i \in [t]|x) = \mathbb{E}_w[w_{z_1} \ldots w_{z_t}|x]$ and $W_{post} = \mathbb{E}_w[w^{\otimes t}|x]$.[1] Therefore, any polynomial $P(w)$ of degree at most $t$ can be represented as $\mathbb{E}_w[P(w)|x] = \beta^\top \text{vec}(W_{post})$ for some $\beta$, where $\text{vec}(W_{post}) \in \mathbb{R}^{K^t}$ is the vectorization of $W_{post}$.

We now are ready to present the robust version of Theorem 3. It shows that if we only find a function $f$ whose loss is at most $\epsilon$ larger than the optimal loss, then a linear transformation of our learned representation can still give a good approximation of the target polynomial within a $O(\epsilon)$ error.

**Theorem 4** (Robust Version). *Consider the general topic model setting as Definition 1, suppose topic matrix $A$ satisfies $\text{rank}(A) = K$ and function $f$ satisfies $L_{reconst}(f) \leq L^*_{reconst} + \epsilon$ for some $\epsilon > 0$. Then, any polynomial $P(w)$ of the posterior $w|x, A$ with degree at most $t$ is approximately linear in $f(x)$, that is there exists a $\theta \in \mathbb{R}^{V^t}$ such that*

$$\mathbb{E}_x\left[\left(\mathbb{E}_w[P(w)|x] - \theta^\top f(x)\right)^2\right] \leq 2\|\beta\|^2 \kappa^{2t}(A^\dagger)\epsilon,$$

*where $\mathbb{E}_w[P(w)|x] = \beta^\top vec(W_{post})$.*

Note that the dependency on $\|\beta\|^2 \kappa^{2t}(A^\dagger)$ is expected, since this is the norm $\|\theta\|^2$ that we would have if $\epsilon = 0$, i.e., the $\theta$ we would have in Theorem 3. Thus, this quantity should be understood as the complexity of the target function for the downstream task. Empirically we show that $\kappa(A^\dagger)$ is small in Section C.6. The proof is deferred to Section A.3.

## 4 GUARANTEES FOR THE CONTRASTIVE OBJECTIVE

In this section, we consider the contrastive objective (2) for self-supervised learning and provide similar provable guarantees on its performance as the reconstruction-based objective.

We use the same approach as Tosh et al. (2020) to construct a representation $g(x)$ based on $f(x, \cdot)$. Given a set of landmark documents $\{l_i\}_{i=1}^m$ with length $|l_i| = t$ as references, the representation is defined as

$$g(x, \{l_i\}_{i=1}^m) = (g(x, l_1), \ldots, g(x, l_m))^\top, \quad g(x, x') = \frac{f(x, x')}{1 - f(x, x')}. \tag{3}$$

The following theorem gives the theoretical guarantee for this representation. Similar to Theorem 3 for reconstruction-based objective, it shows that any polynomial $P(w)$ of degree at most $t$ can be representation by a linear function of the learned representation. Our proof here relies on the observation that having $g(x, \{l_i\}_{i=1}^m)$ for all short landmark documents of length $t$ gives similar information as $f$ for the reconstruction objective. This observation allows us to remove the anchor words assumption needed in (Tosh et al., 2020).

---

[1]To simply the notations, WLOG we assume topics set $\mathcal{K} = [K]$ so that we can use topics $z_i \in [K]$ as indices.

**Theorem 5.** *Consider the general topic model setting as Definition 1, suppose topic matrix $A$ satisfies rank$(A) = K$ and function $f$ minimizes the contrastive objective (2). If we randomly sampled $m = K^t$ different landmark documents $\{l_i\}_{i=1}^m$ and construct $g(x, \{l_i\}_{i=1}^m)$ as (3), then any polynomial $P(w)$ of the posterior $w|x, A$ with degree at most $t$ is linear in $g(x, \{l_i\}_{i=1}^m)$, that is there exists a $\theta \in \mathbb{R}^m$ such that for all documents $x$*

$$\mathbb{E}_w[P(w)|x] = \theta^\top g(x, \{l_i\}_{i=1}^m).$$

In fact, one can use the same set of documents for both representation and downstream task so that we do not need additional landmark documents. We call this case as self-reference. The following corollary shows that solving the downstream task is equivalent to solve a kernel regression (in some sense the landmark documents are just random features for this kernel (Rahimi et al., 2007)). See more discussions in Section B.3.

**Corollary 6.** *Denote $\{(x_i, \tilde{y}_i)\}_{i=1}^m$ as the downstream dataset, where $\tilde{y}_i = \mathbb{E}_w[P(w)|x_i]$ is the target for document $x_i$. If we set landmarks $\{l_i\}_{i=1}^m$ to be $\{x_i\}_{i=1}^m$, then solve the downstream task is equivalent to a kernel regression, that is*

$$\arg\min_\theta \sum_{i=1}^m \left(\tilde{y}_i - \theta^\top g(x, \{l_i\}_{i=1}^m)\right)^2 = \arg\min_\theta \|\tilde{y} - G\theta\|^2,$$

*where $\tilde{y} = (\tilde{y}_1, \ldots, \tilde{y}_m)^\top$ and $G \in \mathbb{R}^{m \times m}$ is a kernel matrix such that $G_{ij} = g(x_i, x_j)$.*

## 5 SYNTHETIC EXPERIMENTS

In this section, we optimize the reconstruction-based objective (1) on data generated by several topic models to show that the objective can adapt to different priors and recover topic posterior accurately.

### 5.1 TOPIC MODELS

We consider four types of topic models in our experiments. Our first topic model is the pure-topic model, where each document's topic comes from a discrete uniform distribution over the $K$-dimensional linear basis, namely $\{e_1, e_2, ..., e_K\}$. Our second topic model is the Latent Dirichlet Allocation (LDA) model, where $\Delta(K)$ is a symmetric Dirichlet distribution $\text{Dir}(1/K)$.

We are also interested in topic models that involve more subtle topic correlations and similarities between different topic's word distributions. To this end, we consider the Correlated Topic Model (CTM) (Blei & Lafferty, 2007) and the Pachinko Allocation Model (PAM) (Li & McCallum, 2006). Our goal is to construct settings where the correlation between topics provide useful information for inference. To achieve this goal, in CTM, we construct groups of 4 topics. Within the group, topic pairs (0,2) and (1,3) are highly correlated (as specified by the prior), while topics 0 and 1 share many words (see Figure 1 for an illustration of the setting). In this case, if we observe a document with words that could either belong to topic 0 or 1, but this same document also has words that are associated with topic 2, we can infer that the first set of words are likely from topic 0, not topic 1. We construct such correlations in CTM model by setting the diagonal entries of its Gaussian covariance matrix to 15 and the covariance between correlated pairs of topics to 0.99 times the diagonal entries, with the remaining entries set to zero. We construct similar examples for Pachinko Allocation Model (PAM), see Section C in appendix for details.

### 5.2 SIMULATION SETUP

**Document generation** Our documents are synthetically generated through the following steps: we first construct a $V \times K$ topic-word matrix $A$. Then, for each document, we determine the document length $n$ from Poisson distribution $\text{Pois}(\lambda)$, draw a topic distribution $w$ from $\Delta(K)$, and draw $n$ words i.i.d. from the word distribution given by $Aw$. In our simulation, we set $K = 20, V = 5000, \lambda \in \{30, 60\}$. Often $A$ will be drawn from a dirichlet distribution $\text{Dir}(\alpha/K)$, and we take $\alpha = 1, 3, 5, 7, 9$ to vary the difficulty level of the inference problem, where a larger $\alpha$ introduces more similarity between topics.

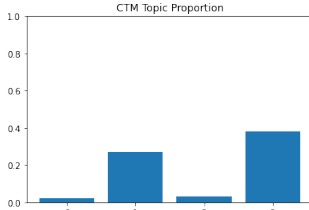 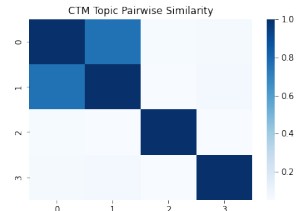

Figure 1: One example of a group of 4 topics in the Correlated Topic Model. Left: Weight of each topic in a document's topic proportion. In this example topics 1 and 3 have large proportions as they are correlated. Right: Pairwise topic similarity.

| TV Distance | Document Type | | | | Major Topic(s) Recovery | Document Type | | | |
|---|---|---|---|---|---|---|---|---|---|
| $\alpha$ | Pure | LDA | CTM | PAM | $\alpha$ | Pure | LDA | CTM | PAM |
| 1 | 0.0148 | 0.0757 | 0.0550 | 0.0489 | 1 | 1.0000 | 0.9050 | 0.9175 | 0.9025 |
| | ± 0.0020 | ± 0.0033 | ± 0.0037 | ± 0.0025 | | ± 0.0000 | ± 0.0406 | ± 0.0275 | ± 0.0330 |
| 3 | 0.0308 | 0.0899 | 0.0799 | 0.0712 | 3 | 1.0000 | 0.8750 | 0.8900 | 0.8950 |
| | ± 0.0076 | ± 0.0053 | ± 0.0048 | ± 0.0038 | | ± 0.0000 | ± 0.0458 | ± 0.0311 | ± 0.0344 |
| 5 | 0.0501 | 0.1041 | 0.0970 | 0.0787 | 5 | 1.0000 | 0.9000 | 0.8975 | 0.8675 |
| | ± 0.0030 | ± 0.0062 | ± 0.0057 | ± 0.0043 | | ± 0.0000 | ± 0.0416 | ± 0.0296 | ± 0.0407 |
| 7 | 0.0391 | 0.1233 | 0.1071 | 0.0960 | 7 | 1.0000 | 0.9150 | 0.8675 | 0.8675 |
| | ± 0.0022 | ± 0.0071 | ± 0.0068 | ± 0.0057 | | ± 0.0000 | ± 0.0387 | ± 0.0383 | ± 0.0407 |
| 9 | 0.0517 | 0.1358 | 0.1101 | 0.0971 | 9 | 0.9884 | 0.9100 | 0.8850 | 0.8325 |
| | ± 0.0045 | ± 0.0089 | ± 0.0064 | ± 0.0053 | | ± 0.0097 | ± 0.0397 | ± 0.0330 | ± 0.0461 |

Table 1: Left: TV distance between recovered topic posterior and true topic posterior for different topic models. Right: Major topic(s) recovery rate for different topic models. The 95% confidence interval is reported in both tables.

**Neural network models**  We transform bag-of-word document representation into full document by repeating each word by its frequency and concatenating them in random order; afterwards, the full document will be our model's input. We find that fully-connected neural network with residual connections performs the best on recovering topic posterior distribution for pure-topic and LDA documents, and attention-based architecture (Vaswani et al., 2017) performs the best for recovering topic posterior distribution for CTM and PAM documents. More specifically, the attention-based model contains 8 transformer blocks where each transformer block consists of a 768-dimensional attention layer and a feed-forward layer, with residual connection applied around every block. The model's second to last layer averages over the outputs of the last transformer block and the final layer projects it to a $V$-dimensional word distribution.

**Training setup**  During training, we resample 60,000 new documents after every 2 epochs, and the total amount of training data varies from 720K documents to 6M documents. The loss function is the reconstruction-based objective (1) with $t = 1$, i.e., we want the model to predict one missing word. To reduce the variance during the training, in each training document, the last 6 words are chosen as the prediction target and hidden from the model. The trained model is evaluated on test documents generated from the same topic prior as the training documents. We use 5,000 test documents for the pure-topic prior and 200 test documents for the remaining priors. For each test document, we use a Markov Chain Monte Carlo No U-Turn Sampler (Salvatier et al., 2016) assuming the document's correct topic prior to approximate the ground truth expected topic posterior distribution. Then, we measure topic posterior recovery loss as the Total Variation (TV) distance between the recovered topic posterior and the ground truth topic posterior.

## 5.3 RESULTS

**Topic posterior recovery loss**  As illustrated in Table 1, after 200 training epochs, our model can accurately recover the topic posterior distribution. Meanwhile, it can be observed that for larger values of the Dirichlet hyperparameter $\alpha$, the recovery loss gets higher. This is expected because higher $\alpha$ leads to more similar topics, which makes the learning problem more difficult. This effect is also captured by Definition 3 and we show the computed condition numbers in Section C.6.

| TV Distance | Document Type ($\alpha = 1$) | | | | Major Topic(s) Recovery | Document Type ($\alpha = 1$) | | |
|---|---|---|---|---|---|---|---|---|
| Method | Pure | LDA | CTM | PAM | Method | LDA | CTM | PAM |
| LDA | 0.0406 ± 0.0016 | - | 0.1182 ± 0.0099 | 0.1218 ± 0.0096 | LDA | **0.9150** **± 0.0387** | 0.8850 ± 0.0344 | 0.8500 ± 0.0360 |
| CTM | 0.2083 ± 0.0038 | 0.2060 ± 0.0064 | - | 0.3154 ± 0.0082 | CTM | 0.9000 ± 0.0416 | **0.9175** **± 0.0275** | 0.8300 ± 0.0370 |
| PAM | 0.3782 ± 0.0038 | 0.3459 ± 0.0128 | 0.3939 ± 0.0096 | - | PAM | 0.8750 ± 0.0458 | 0.7250 ± 0.0397 | **0.9050** **± 0.0320** |
| **SSL (ours)** | **0.0148** **± 0.0020** | **0.0757** **± 0.0033** | **0.0550** **± 0.0037** | **0.0489** **± 0.0025** | **SSL (ours)** | 0.9050 ± 0.0406 | **0.9175** **± 0.0275** | 0.9025 ± 0.0330 |

Table 2: Left: TV distance between recovered topic posterior and true topic posterior of our self-supervised learning approach versus posterior inference via Markov Chain Monte Carlo assuming a specific prior for $\alpha = 1$. Right: Major topic recovery rate of our approach versus posterior inference via Markov Chain Monte Carlo assuming a specific prior for $\alpha = 1$. In both the Left and Right table, the 95% confidence interval is reported.

.

**Major topic recovery** We examine the extent our recovered topic posterior captures the major topics in a given document. For pure-topic and LDA documents, we measure major topic recovery probability of correctly estimating the topic with largest proportion. Considering that topics in CTM and PAM documents are correlated in pairs, we measure the major topic recovery rate for CTM and PAM as the top-2 topic overlap rate on their test documents. Table 1 shows that our algorithm is successful throughout all settings we've tried.

**Robustness of self-supervised learning** We compare the self-supervised approach to traditional topic inference. For each $\alpha$ value, we take 200 test documents from each category of documents, and we run posterior inference assuming a specific topic model and calculate the TV distance between this posterior and the ground truth posterior. We exclude assuming pure-topic prior from our comparison because it gives invalid results for documents with mixed topics. For $\alpha = 1$, as shown in Table 2 Left, the topic posterior recovered from our approach is closer to the ground truth topic posterior than that recovered from a mis-specified topic prior.

We also test the major topic recovery rate for our model against topic inference using both correct and incorrect model (see Table 2 Right). The major topic recovery rate of our approach is similar to that of posterior inference assuming the correct prior. For pure topic model, all four methods get 100% major topics recovery rate on pure-topic documents; for LDA, again all four methods perform similarly well since the instance is not difficult. However we observe significant difference for more complicated CTM and PAM models. For these models, the self-supervised learning approach performs similarly to posterior inference using the correct model, and both of them are significantly better than posterior inference using mis-specified models. The results for $\alpha = 3, 5, 7, 9$ are presented in Section C. The comparison further reveals the robustness of self-superivised learning.

## 6 EXPERIMENTS ON REAL DATA

Even though our theory suggests that self-supervised learning can learn useful representations for a general topic model, it is still unclear whether this simple approach learns any reasonable representation in practice as the documents are not really generated by *any* topic model. In this section, we show that the simple approach at least generates a better representation than several baselines.

### 6.1 EXPERIMENT SETUP

The dataset we used is the AG news dataset (Zhang et al., 2015), in which each document has a label of one out of four categories: world, sports, business, and sci/tech. Thus, the task falls into pure topic model. We generally follow experiment setup done by Tosh et al. (2020) (see more details in Section D.1), but the representation we used is generated by our reconstruction-based objective.

Our training involves two major phases: unsupervised phase and supervised phase. In the unsupervised phase, we trained our self-supervised model on most of our data that is later used for generating document representations; and then in the supervised phase, we trained a linear classifier

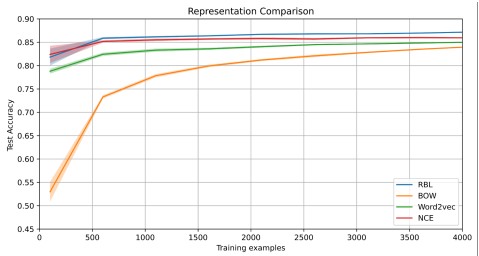

Figure 2: Representation Comparison RBL, NCE with baselines of BOW and Word2vec.

using multi-class Logistics Regression, implemented by Scikit-learn(Pedregosa et al. (2011)), with 3-fold Cross Validation for parameter tuning ($l_2$ regularization term and solver).

We chose residual blocks as the basic building block for our neural network architecture, with varying width and depth. The network used for Figure 2 has 3 residual blocks and a width of 4096 neurons per layer. We explore other depths/weight combinations in Section D.3 in Appendix. We used AMSGrad optimizer (Reddi et al., 2019) with weight decay of 0.01. We resample our reconstruction-based document-label pair every 2 epochs. We trained the model using reconstruction-based objective (1) with $t = 1$. Using the same variance-reduction technique in synthetic experiments, in actual training, we sampled 4 words as our labels and the loss is an average of the cross entropy loss of each word evaluated separately.

To extract our representation, we use an identity matrix to replace the last layer. That is, we take a softmax function on top of the second-to-last layer of the neural network. This is different from our theory but we found that it effectively reduces the high output dimension (equal to vocabulary size, around 16,700 in our experiments) and improves performance. We included more details in Section D.2 in Appendix.

## 6.2 EXPERIMENT RESULT

The performance of our Reconstruction-based learning representation (RBL) is shown in Figure 2, where the test accuracy on representation is plotted against number of training samples used to train the classifier in supervised learning phase. Our representation RBL performs better than both BOW and Word2vec baselines. Notably, Word2vec representation is inferior only by a small margin, and works well in general even when limited training samples are provided. On the contrary, although BOW representation has a decent test accuracy when training samples are abundant, it performs significantly poorly on smaller training set.

We also compared our result to previous noise contrastive estimation (NCE) representation benchmark achieved by Tosh et al. (2020) using contrastive objective , where their reported best accuracy was around $87.5\%$ when full 4000 training samples are used, slightly higher than our RBL corresponding test accuracy of $87.1\%$. In our attempt to reproduce their result, parameter tuning yields the best accuracy of $86\%$ when full training samples were used. We plot our own reproduced results on in Figure 2 since parameter tuning did not achieve a close benchmark to their result.

## 7 CONCLUSION

"All models are wrong but some are useful." If one self-supervised objective can capture all models, then it would be able to extract useful information. In this paper, we studied the self-supervised learning in the topic models setup and showed that it can provide useful information about the topic posterior no matter what topic model is used. Our results generalized previous work (Tosh et al., 2020) to both contrastive learning and reconstruction-based learning and our techniques allow us to depend on weaker assumptions. We also empirically showed that the reconstruction-based learning performs better than the posterior inference under mis-specified models, and it can provide useful representation for the topic inference problem. An immediate open problem is what other models (especially those that captures ordering of words) can be captured by the much more general self-supervised learning approaches used in practice.

## REPRODUCIBILITY STATEMENT

For our theoretical results, we explain the detailed setup in Section 2, give the formal statement of the results in Section 3 and Section 4, and present the complete proofs in Section A and Section B. For our experiments, we report our results and general setup in Section 5 and Section 6 and explain detailed setup in Section C and Section D. We also include our codes that generate the results in the supplement material.

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

## A  OMITTED PROOFS IN SECTION 3

In this section, we give the omitted proofs in Section 3. We first give a warmup example to illustrate our proof idea in Section A.1. Then we give the proof of our main result (Theorem 3) in Section A.2 and the proof of robust version (Theorem 4) in Section A.3.

### A.1  WARM-UP EXAMPLE: RECONSTRUCT THE UNKNOWN WORD

In this warm-up example, we consider the simple setting where we try to predict the only one unknown word of the document using reconstruction-based objective (1). The following result is the special case of Theorem 3 with $t = 1$, which shows that the learned representation is able to give any linear function of the topic posterior.

**Theorem A.1.** *Consider the general topic model setting as Definition 1, suppose topic matrix $A$ satisfies rank$(A) = K$ and function $f$ minimizes the reconstruction-based objective* (1). *Then, any linear function $P(w)$ of $w$ is linear in $f(x)$, that is for any $P(w) = \beta^\top w$, there exists a $\theta \in \mathbb{R}^{\check{V}}$ such that for all documents $x$*

$$\mathbb{E}_w \left[ \beta^\top w | x \right] = \theta^\top f(x).$$

Following the proof idea described in Section 3, we first give a characterization of the optimal function $f$. The following lemma shows that $f(x)$ must be the word posterior vector given the document $x$. The proof of this lemma is simply based on the property of the cross-entropy loss function and we defer it to Section A.1.1. Recall that our vocabulary set is $\mathcal{V} = \{v_1, \ldots, v_V\}$ and $x_{\text{unsup}} = (x, y)$ is the given document, where $x$ is unmasked part and $y$ is word marked as unknown.

**Lemma A.2.** *If $f$ minimizes the reconstruction-based objective* (1)*, then we have for all document $x$*

$$f(x) = (\mathbb{P}(y = v_1 | x), \ldots, \mathbb{P}(y = v_V | x))^\top.$$

Based on the above lemma, we are able to prove Theorem A.1. It is easy to see that the word posterior distribution is $A\mathbb{E}_w[w|x]$, so we have $f(x) = A\mathbb{E}_w[w|x]$. Since the columns of $A$ are linearly independent, we know $\mathbb{E}_w[w|x] = A^\dagger f(x)$. Thus, for any linear function $\beta^\top w$, there exists $\theta = (A^\dagger)^\top \beta$ such that $\mathbb{E}_w[\beta^\top w|x] = \theta^\top f(x)$. The formal proof of the general case is given in Section A.2.

#### A.1.1  PROOF OF LEMMA A.2

Instead of focusing on the $t = 1$ case, we directly give the corresponding lemma of Lemma A.2 for general $t$. Recall that $x_{\text{unsup}} = (x, y)$ is the given document, $x$ is the unmasked part, $y = (y_1, \ldots, y_t)$ is the $t$ unknown words that we want to predict and $\mathcal{V} = \{v_1, \ldots, v_V\}$ is the set of vocabulary. It shows that the optimal $f$ is the $t$ words posterior distribution. Note that the words posterior distribution is linear in the topic posterior of $t$-th moment, which is useful for the later analysis.

**Lemma A.3** (Lemma A.2, General Case). *If $f$ minimizes the reconstruction-based objective* (1)*, then we have for all document $x$*

$$f(x) = (\mathbb{P}(y = (v_1, v_1, \ldots, v_1)|x), \mathbb{P}(y = (v_2, v_1, \ldots, v_1)|x), \ldots, \mathbb{P}(y = (v_V, v_V, \ldots, v_V)|x))^\top.$$

*Proof.* Denote the word posterior distribution as $p^*(x)$. We will show $f(x) = p^*(x)$. By the law of total expectation, we have

$$L_{\text{reconst}} = \mathbb{E}_{x,y}[\ell(f(x), y)] = \mathbb{E}_x[\mathbb{E}_{y|x}[\ell(f(x), y)|x]].$$

We know the probability of $y$ given $x$ is $p^*(x)$. Since $\ell$ is cross-entropy loss, we have

$$L_{\text{reconst}} \geq \mathbb{E}_x \left[ - \sum_{k \in [V^t]} [p^*(x)]_k \log[p^*(x)]_k \right],$$

where the equality is obtained when $f(x) = p^*(x)$. Thus, when $f$ minimizes the reconstruction-based objective (1), we have $f(x) = p^*(x)$. □

A.2    PROOF OF THEOREM 3

In this section, we give the proof of Theorem 3. Theorem A.1 is covered by Theorem 3 as the special case of $t = 1$. The proof follows the same idea as for Theorem A.1. We first show that $f(x)$ is the $t$-words posterior (Lemma A.3). Then just as $t = 1$ case where we can recover topic posterior with $A^\dagger f(x)$, we can also recover topic posterior of $t$-th moment with $\mathcal{A}^\dagger f(x)$ with some matrix $\mathcal{A}$. The result follows by the observation that for a polynomial $P(w)$ with degree at most $t$, $\mathbb{E}_w[P(w)|x]$ is a linear function of the topic posterior of $t$-th moment.

**Theorem 3** (Main Result). *Consider the general topic model setting as Definition 1, suppose topic matrix $A$ satisfies rank$(A) = K$ and function $f$ minimizes the reconstruction-based objective* (1). *Then, any polynomial $P(w)$ of the posterior $w|x, A$ with degree at most $t$ is linear in $f(x)$, that is there exists a $\theta \in \mathbb{R}^{V^t}$ such that for all documents $x$*

$$\mathbb{E}_w[P(w)|x] = \theta^\top f(x).$$

*Proof.* Since $f$ is the minimizer of reconstruction-based objective (1), by Lemma A.3 we know given an input document $x$,

$$f(x) = \left(\mathbb{P}(y = (v_1, v_1, \ldots, v_1)|x), \mathbb{P}(y = (v_2, v_1, \ldots, v_1)|x), \ldots, \mathbb{P}(y = (v_V, v_V, \ldots, v_V)|x)\right)^\top.$$

We will show that $f(x)$ is linear in the topic posterior. In the following, we focus on $[f(x)]_{y_1, \ldots, y_t}$, which is word posterior probability of the $t$ unknown words being $y_1, \ldots, y_t \in \mathcal{V}$ given document $x$. Recall $\Delta^{K-1}$ be the $K - 1$ dimensional probability simplex. By the law of total probability, we have

$$
\begin{aligned}
[f(x)]_{y_1, y_2, \ldots, y_t} &= \mathbb{P}(y_1, y_2, \ldots, y_t|x) \\
&= \int_{w \in \Delta^{K-1}} \mathbb{P}(y_1, y_2, \ldots, y_t, w|x)\mathrm{d}w \\
&= \int_{w \in \Delta^{K-1}} \mathbb{P}(y_1, y_2, \ldots, y_t|w, x)\mathbb{P}(w|x)\mathrm{d}w \\
&= \int_{w \in \Delta^{K-1}} \sum_{z_1, \ldots, z_t \in [K]} \mathbb{P}(y_1, y_2, \ldots, y_t, z_1, z_2, \ldots z_t|w)\mathbb{P}(w|x)\mathrm{d}w \\
&= \int_{w \in \Delta^{K-1}} \sum_{z_1, \ldots, z_t \in [K]} \mathbb{P}(z_1, , \ldots, z_t|w)\mathbb{P}(y_1, \ldots, y_t|z_1, \ldots, z_t, w)\mathbb{P}(w|x)\mathrm{d}w.
\end{aligned}
$$

Note that $z_i$ is the topic of word $y_i$. Since we consider the general topic model as Definition 1, we know

$$\mathbb{P}(y_1, y_2, \ldots, y_t|z_1, z_2, \ldots, z_t, w) = \mathbb{P}(y_1, y_2, \ldots, y_t|z_1, z_2, \ldots, z_t) = \prod_{i=1}^{t} \mathbb{P}(y_i|z_i) = \prod_{i=1}^{t} A_{y_i, z_i},$$

where $A$ is the topic matrix. Hence,

$$
\begin{aligned}
[f(x)]_{y_1, y_2, \ldots, y_t} &= \sum_{z_1, z_2, \ldots z_t \in [K]} \prod_{i=1}^{t} A_{y_i, z_i} \int_{w \in \Delta^{K-1}} \mathbb{P}(z_1, z_2, \ldots, z_t|w)\mathbb{P}(w|x)\mathrm{d}w \\
&= \sum_{z_1, z_2, \ldots z_t \in [K]} \prod_{i=1}^{t} A_{y_i, z_i} \int_{w \in \Delta^{K-1}} \prod_{i=1}^{t} w_{z_i}\mathbb{P}(w|x)\mathrm{d}w \\
&= \sum_{z_1, z_2, \ldots z_t \in [K]} \prod_{i=1}^{t} A_{x_{m+i}, z_i} \mathbb{E}_w\left[\prod_{i=1}^{t} w_{z_i}\Big|x\right].
\end{aligned}
$$

Recall that the topic posterior tensor is $W_{post} = \mathbb{E}_w[w^{\otimes t}|x] \in \mathbb{R}^{K \times \ldots \times K}$, where each entry $[W_{post}]_{z_1,\ldots,z_k} = \mathbb{P}(z_i \text{ is the topic of word } y_i \text{ for } i \in [t]|x) = \mathbb{E}_w[w_{z_1} \ldots w_{z_t}|x]$. Therefore,

$$[f(x)]_{y_1,y_2,\ldots,y_t} = \sum_{z_1,z_2,\ldots z_t \in [K]} \prod_{i=1}^{t} A_{y_i,z_i} [W_{post}]_{z_1,z_2,\ldots,z_t},$$

$$f(x) = (\underbrace{A \otimes A \otimes \cdots \otimes A}_{t \text{ times}}) \text{vec}(W_{post}),$$

where $\otimes$ is the Kronecker product, $\mathcal{A} = A \otimes A \otimes \cdots \otimes A \in \mathbb{R}^{V^t \times K^t}$ and $\text{vec}(W_{post}) \in \mathbb{R}^{K^t}$ is the vectorization of $W_{post}$. Note that $\mathcal{A}^\dagger = A^\dagger \otimes \cdots \otimes A^\dagger$, so we have $\text{vec}(W_{post}) = \mathcal{A}^\dagger f(x)$.

Since $P(w)$ is a polynomial of degree at most $t$, we know there exists $\beta$ such that $\mathbb{E}_w[P(w)|x] = \beta^\top \text{vec}(W_{post})$. Thus, let $\theta = (\mathcal{A}^\dagger)^\top \beta$, we have $\mathbb{E}_w[P(w)|x] = \beta^\top \text{vec}(W_{post}) = \theta^\top f(x)$. $\qquad \square$

### A.3 PROOF OF THEOREM 4

Similar to Lemma A.2, we can show that the representation $f(x)$ is close to $t$ words posterior. Then, the key lemma to obtain Theorem 4 is the following result. It suggests that if function $f$ is $\epsilon$-close to the optimal function $f^*$ (word posterior distribution) under cross-entropy loss, then we can still recover the topic posterior with proper $B$ up to $O(\sqrt{\epsilon})$ error. For example, we will choose $B = A^\dagger$ when $t = 1$. See the proof of Theorem 4 for the choice of $B$ in the general case.

**Lemma A.4.** *For cross-entropy loss $\ell$ and any two probability vectors $p, p^*$, if $\ell(p, p^*) \leq \min_{p:\sum_i p_i=1,p \geq 0} \ell(p, p^*) + \epsilon$, then for any matrix $B$ we have $\|Bp - Bp^*\|_1 \leq \kappa(B)\sqrt{2\epsilon}$.*

*Proof.* Recall that $\ell$ is cross-entropy loss, we know

$$\ell(p, p^*) - \min_{p:\sum_i p_i=1,p \geq 0} \ell(p, p^*) = D_{KL}(p \,||\, p^*) \leq \epsilon.$$

By Pinsker's Inequality, we have that

$$\|p - p^*\|_1 \leq \sqrt{2D_{KL}(p^* \,||\, p)} \leq \sqrt{2\epsilon}.$$

Then, for any matrix $B$ we have

$$\|Bp - Bp^*\|_1 \leq \kappa(B)\|p - p^*\|_1 \leq \kappa(B)\sqrt{2\epsilon},$$

where we use the definition of $\ell_1$ condition number (Definition 3). $\qquad \square$

We now are ready to give the proof of Theorem 4, which is based on Lemma A.4 and the proof of Theorem 3.

**Theorem 4** (Robust Version). *Consider the general topic model setting as Definition 1, suppose topic matrix $A$ satisfies $\text{rank}(A) = K$ and function $f$ satisfies $L_{reconst}(f) \leq L^*_{reconst} + \epsilon$ for some $\epsilon > 0$. Then, any polynomial $P(w)$ of the posterior $w|x, A$ with degree at most $t$ is approximately linear in $f(x)$, that is there exists a $\theta \in \mathbb{R}^{V^t}$ such that*

$$\mathbb{E}_x \left[ \left( \mathbb{E}_w[P(w)|x] - \theta^\top f(x) \right)^2 \right] \leq 2 \|\beta\|^2 \kappa^{2t}(A^\dagger)\epsilon,$$

*where $\mathbb{E}_w[P(w)|x] = \beta^\top vec(W_{post})$.*

*Proof.* Recall from the proof of Theorem 3 that $\mathbb{E}_w[P(w)|x] = \beta^\top \text{vec}(W_{post}) = \beta^\top \mathcal{A}^\dagger f^*(x)$, where $\mathcal{A}^\dagger = A^\dagger \otimes \cdots \otimes A^\dagger$ and $\kappa(\mathcal{A}^\dagger) = \kappa^t(A^\dagger)$. Same as in the proof of Theorem 3, let $\theta = (\mathcal{A}^\dagger)^\top \beta$. We have

$$\mathbb{E}_x \left[ \left( \mathbb{E}_w[P(w)|x] - \theta^\top f(x) \right)^2 \right] = \mathbb{E}_x \left[ \left( \beta^\top \mathcal{A}^\dagger f^*(x) - \beta^\top \mathcal{A}^\dagger f(x) \right)^2 \right]$$

$$\leq \mathbb{E}_x \left[ \|\beta\|_2^2 \left\| \mathcal{A}^\dagger f^*(x) - \mathcal{A}^\dagger f(x) \right\|_2^2 \right]$$

$$\leq \|\beta\|_2^2 \, \mathbb{E}_x \left[ \left\| \mathcal{A}^\dagger f^*(x) - \mathcal{A}^\dagger f(x) \right\|_1^2 \right].$$

We are going to use Lemma A.4 to bound $\left\|\mathcal{A}^\dagger f^*(x) - \mathcal{A}^\dagger f(x)\right\|_1$. By Lemma A.3 we know $f^*(x)$ is the word posterior, so $f^*(x) = \mathbb{E}_{y|x}[y|x]$. Since $\ell$ is cross-entropy loss, we know

$$\mathbb{E}_{y|x}[\ell(f(x), y) - \ell(f^*(x), y)|x] = \mathbb{E}_{y|x}\left[\sum_{k \in [V^t]} y_k \log\left(\frac{[f^*(x)]_k}{[f(x)]_k}\right)\bigg|x\right] = D_{KL}(f^*(x) \,||\, f(x)),$$

which implies $\ell(f(x), f^*(x)) - \min_p \ell(p, f^*(x)) \leq \epsilon$. Therefore, by Lemma A.4 with $B = \mathcal{A}^\dagger$, we know

$$\left\|\mathcal{A}^\dagger f(x) - \mathcal{A}^\dagger f^*(x)\right\|_1 \leq \kappa(\mathcal{A}^\dagger)\sqrt{2\epsilon} = \kappa^t(A^\dagger)\sqrt{2\epsilon}.$$

Given the desired bound, we have

$$\mathbb{E}_x\left[\left(\mathbb{E}_w[P(w)|x] - \theta^\top f(x)\right)^2\right] \leq 2\left\|\beta\right\|_2^2 \kappa^{2t}(A^\dagger)\epsilon.$$

$\square$

# B  OMITTED PROOFS IN SECTION 4

In this section, we give the omitted proofs in Section 4. We give the proof of Theorem 5 in Section B.1 and proof of Lemma B.1 in Section B.2.

## B.1  PROOF OF THEOREM 5

Before presenting the proof of Theorem 5, we first give a characterization of the representation $g(x, x')$, which would be useful in the later analysis. Note that this the same as the one shown in Tosh et al. (2020). We provide its proof for completeness in Section B.2.

**Lemma B.1.** *If $f$ minimizes the contrastive objective* (2)*, then we have*

$$g(x, x') \triangleq \frac{f(x, x')}{1 - f(x, x')} = \frac{\mathbb{P}(y = 1 | x, x')}{\mathbb{P}(y = 0 | x, x')}.$$

Now we are ready to proof Theorem 5.

**Theorem 5.** *Consider the general topic model setting as Definition 1, suppose topic matrix $A$ satisfies rank$(A) = K$ and function $f$ minimizes the contrastive objective* (2)*. If we randomly sampled $m = K^t$ different landmark documents $\{l_i\}_{i=1}^m$ and construct $g(x, \{l_i\}_{i=1}^m)$ as* (3)*, then any polynomial $P(w)$ of the posterior $w|x, A$ with degree at most $t$ is linear in $g(x, \{l_i\}_{i=1}^m)$, that is there exists a $\theta \in \mathbb{R}^m$ such that for all documents $x$*

$$\mathbb{E}_w[P(w)|x] = \theta^\top g(x, \{l_i\}_{i=1}^m).$$

*Proof.* Since $f$ is the minimizer of contrastive objective (2), by Lemma B.1 we know

$$g(x, x') = \frac{\mathbb{P}(y = 1 | x, x')}{\mathbb{P}(y = 0 | x, x')}.$$

Similar to the proof of Theorem 3, we will show that $g(x, x')$ is linear in the topic posterior when $x'$ is fixed. By Bayes' rule, we have

$$g(x, x') = \frac{\mathbb{P}(x, x'|y = 1)\mathbb{P}(y = 1)/\mathbb{P}(x, x')}{\mathbb{P}(x, x'|y = 0)\mathbb{P}(y = 0)/\mathbb{P}(x, x')} = \frac{\mathbb{P}(x, x'|y = 1)}{\mathbb{P}(x, x'|y = 0)},$$

where we use $\mathbb{P}(y = 0) = \mathbb{P}(y = 1) = 1/2$.

Denote the $t$ words in $x'$ as $x'_1, \ldots, x'_t$ and their corresponding topics as $z_1, \ldots, z_t$. Then, by our way of generating $x, x'$, we know

$$\mathbb{P}(x, x'|y = 0) = \mathbb{P}(x)\mathbb{P}(x'),$$

$$\mathbb{P}(x, x'|y = 1) = \int_w \mathbb{P}(x'_1, \ldots, x'_t|w, x, y = 1)\mathbb{P}(w, x|y = 1)\mathrm{d}w$$

$$= \int_w \sum_{z_1, \ldots, z_t \in [K]} \mathbb{P}(x'_1, \ldots, x'_t|z_1, \ldots, z_t, w)\mathbb{P}(z_1, \ldots, z_t|w)\mathbb{P}(w, x)\mathrm{d}w,$$

which implies

$$g(x, x') = \frac{1}{\mathbb{P}(x')} \int_w \sum_{z_1, \ldots, z_t \in [K]} \mathbb{P}(x'_1, \ldots, x'_t|z_1, \ldots, z_t)\mathbb{P}(z_1, \ldots, z_t|w)\mathbb{P}(w|x)\mathrm{d}w$$

Note that

$$\mathbb{P}(x'_1, \ldots, x'_t|z_1, \ldots, z_t) = \prod_{i=1}^t \mathbb{P}(x'_i|z_i) = \prod_{i=1}^t A_{x'_1, z_i},$$

where $A$ is the topic matrix. Hence,

$$
\begin{aligned}
g(x, x') &= \frac{1}{\mathbb{P}(x')} \sum_{z_1,\ldots,z_t \in [K]} \prod_{i=1}^{t} A_{x_1', z_i} \int_w \mathbb{P}(z_1, \ldots, z_t | w) \mathbb{P}(w | x) \mathrm{d}w \\
&= \frac{1}{\mathbb{P}(x')} \sum_{z_1,\ldots,z_t \in [K]} \prod_{i=1}^{t} A_{x_1', z_i} \int_w \prod_{i=1}^{t} w_{z_i} \mathbb{P}(w | x) \mathrm{d}w \\
&= \frac{1}{\mathbb{P}(x')} \sum_{z_1,\ldots,z_t \in [K]} \prod_{i=1}^{t} A_{x_1', z_i} \mathbb{E}_w \left[ \prod_{i=1}^{t} w_{z_i} \middle| x \right].
\end{aligned}
$$

Recall that the topic posterior tensor is $W_{post} = \mathbb{E}_w[w^{\otimes t} | x] \in \mathbb{R}^{K \times \ldots \times K}$, where each entry $[W_{post}]_{z_1,\ldots,z_k} = \mathbb{E}_w[w_{z_1} \ldots w_{z_t} | x]$. Therefore,

$$
\begin{aligned}
g(x, x') &= \frac{1}{\mathbb{P}(x')} \sum_{z_1,z_2,\ldots z_t \in [K]} \prod_{i=1}^{t} A_{x_i', z_i} [W_{post}]_{z_1,z_2,\ldots,z_t} \\
&= \frac{1}{\mathbb{P}(x')} \mathcal{A}[x']^\top \mathrm{vec}(W_{post}),
\end{aligned}
$$

where $\mathcal{A} = A \otimes A \otimes \cdots \otimes A \in \mathbb{R}^{V^t \times K^t}$, $\mathcal{A}[x'] \in \mathbb{R}^{K^t}$ is the row of $\mathcal{A}$ that correspond to $x_1', \ldots, x_t'$, and $\mathrm{vec}(W_{post}) \in \mathbb{R}^{K^t}$ is the vectorization of $W_{post}$.

Recall we have $m$ randomly sampled different documents $\{l_i\}_{i=1}^n$. Denote $D \in \mathbb{R}^{m \times m}$ as a diagonal matrix such that $D_{i,i} = \mathbb{P}(l_i)$, and $\tilde{\mathcal{A}} \in \mathbb{R}^{m \times K^t}$ as a submatrix of $\mathcal{A}$ such that each row of $\tilde{\mathcal{A}}$ is $\mathcal{A}[l_i]$. Since $l_i$ are randomly sampled, so $D_{i,i} > 0$. Thus, we know

$$
g(x, \{l_i\}_{i=1}^n) = D^{-1} \tilde{\mathcal{A}} \mathrm{vec}(W_{post}),
$$

which implies $\mathrm{vec}(W_{post}) = \tilde{\mathcal{A}}^\dagger D g(x, \{l_i\}_{i=1}^n)$. Note that we need to show $\tilde{\mathcal{A}}^\dagger$ is well-defined. Since $A$ has full column rank, we know $\mathcal{A}$ also has full column rank. Thus, the submatrix $\tilde{\mathcal{A}}$ also has full column rank since $m = K^t$. This implies $\tilde{\mathcal{A}}^\dagger$ is well-defined.

Since $P(w)$ is a polynomial of degree at most $t$, we know there exists $\beta$ such that $\mathbb{E}_w[P(w) | x] = \beta^\top \mathrm{vec}(W_{post})$. Thus, let $\theta = D(\tilde{\mathcal{A}}^\dagger)^\top \beta$, we have

$$
\mathbb{E}_w[P(w) | x] = \beta^\top \mathrm{vec}(W_{post}) = \theta^\top g(x, \{l_i\}_{i=1}^n).
$$

$\square$

## B.2 Proof of Lemma B.1

**Lemma B.1.** *If $f$ minimizes the contrastive objective* (2)*, then we have*

$$
g(x, x') \triangleq \frac{f(x, x')}{1 - f(x, x')} = \frac{\mathbb{P}(y = 1 | x, x')}{\mathbb{P}(y = 0 | x, x')}.
$$

*Proof.* Since $f$ is the minimizer of contrastive objective (2) and

$$
L_{\text{contrast}}(f) = \mathbb{E}_{x,x',y}\left[(f(x, x') - y)^2\right] = \mathbb{E}_{x,x'}\left[\mathbb{E}_{y | x,x'}\left[(f(x, x') - y)^2\right]\right],
$$

it is easy to see that $f$ is the Bayes optimal predictor $\mathbb{P}(y = 1 | x, x')$. Therefore, we know

$$
g(x, x') = \frac{\mathbb{P}(y = 1 | x, x')}{\mathbb{P}(y = 0 | x, x')}.
$$

$\square$

### B.3 DISCUSSIONS ON THE SELF-REFERENCING

As discussed in Corollary 6, we show that one can use the same set of documents for both representation and downstream task. Recall that in Theorem 5 we need a large number of landmark documents to generate the representation and then use it for the downstream task (e.g., fit a polynomial of topic posterior). However, one may not have enough additional documents to be used as landmarks. Therefore, the benefit of the self-referencing is that we do not need additional landmark documents to generate the representation.

The key observation in Corollary 6 is that the learned representation $g(x, x')$ can be viewed as a kernel. Suppose the documents for the downstream task are $x_1, \ldots, x_m$. Construct a kernel matrix $G \in \mathbb{R}^{m \times m}$ such that $G_{ij} = g(x_i, x_j)$. Then, it is easy to see

$$\theta^\top g(x_i, \{x_j\}_{j=1}^m) = \theta^\top G_i,$$

where $G_i$ is the $i$-th column of $G$. Thus, $\theta$ can be found by the following kernel regression:

$$\min_\theta \|\tilde{y} - G\theta\|^2,$$

where $\tilde{y} = (\tilde{y}_1, \ldots, \tilde{y}_m)^\top$ and $\tilde{y}_i = \mathbb{E}_w[P(w)|x_i]$.

# C   SYNTHETIC EXPERIMENT ADDITIONAL DETAILS

In this section, we include more details about our synthetic experiments in Section 5. In Section C.1, we describe the document generation for PAM model. Additional results on topic posterior recovery loss and major topic recovery are included in Section C.2 and Section C.3. In Section C.4 and Section C.5, we report the results about different training epochs and hyperparameters tuning. $\ell_1$ condition number is reported in Section C.6 and we give some visualizations for topic correlations between models in Section C.7.

## C.1   PACHINKO ALLOCATION MODEL TOPIC PRIOR GENERATION

The topic proportion of the Pachinko Allocation Model (PAM) described in Section 5 is generated from the following process: for each document, we first sample a "super-topic" proportion from a symmetric Dirichlet distribution $\text{Dir}(1/K_s)$ and a super-topic-to-topic proportion from a symmetric Dirichlet distribution $\text{Dir}(30)$. Then, we sample each word by first sample a super-topic according to the super-topic proportion and then sample the actual topic from the super-topic-to-topic proportion. In our experiment, we set $K_s = 10$. Figure C.1 offers a few examples of PAM topic proportion generated from the PAM model.

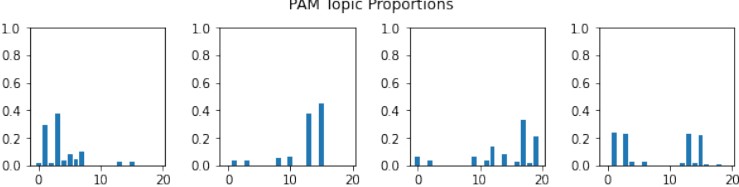

Figure C.1: PAM topic proportion

## C.2   MAJOR TOPIC RECOVERY ADDITIONAL RESULTS

In Section 5.3, we measure the major topics recovery of CTM and PAM documents as the top-2 topic overlap rate. Since topics are correlated in pairs in both of these topic models, we are also interested in the top-4 and top-6 topics overlap rate. As shown in Figure C.2, for both CTM and PAM documents, the average top-4 overlap rate is above 66% and the average top-6 overlap rate is above 62% on all $\alpha$ values we test on. These results show that our model can accurately capture more than just the major pair of correlated topics, but also some correlated pairs of less dominant topics.

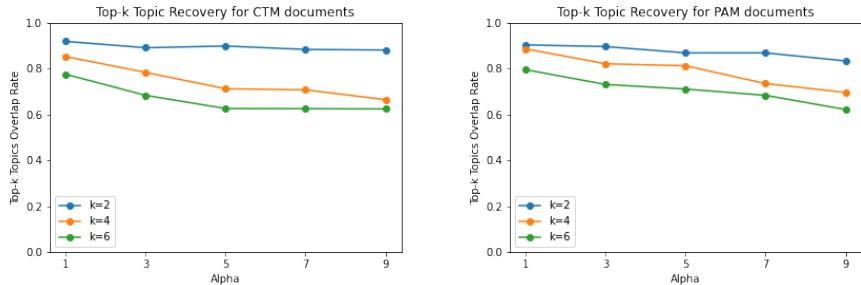

Figure C.2: Recovery rate for top-2, top-4, and top-6 topics, measured on CTM (left) and PAM (right) test documents.

## C.3   SELF-SUPERVISED LEARNING VERSUS ASSUMING SPECIFIC TOPIC PRIOR

We present in Section 5.3 a comparison between the performance of our model and Markov Chain Monte Carlo assuming a specific topic model for $\alpha = 1$, on the task of TV distance and major topic

recovery. In this subsection, we supplement the experiment results for larger values of $\alpha$ along with $\alpha = 1$. In particular, we consider self-supervised learning with 6 million training documents and with 120 thousand training documents after 200 training epochs, and we expect that the former can allow our model to approximate the optimal predictor $\hat{f}$ described in Theorem 3 and the latter may offer some insights on our approach's performance in more practical settings, such as on a real world dataset with limited size. Additionally, when training with 120 thousand documents, we randomly shuffle each document and reassign the $t$ target words in every epoch, to make sure our model learns enough information from the limited amount of training data. We report our results in the form of 95% confidence interval.

| | Document Type ($\alpha = 1$) | | | |
|---|---|---|---|---|
| Method | Pure | LDA | CTM | PAM |
| LDA | 0.0406 ± 0.0016 | - | 0.1182 ± 0.0099 | 0.1218 ± 0.0096 |
| CTM | 0.2083 ± 0.0038 | 0.2060 ± 0.0064 | - | 0.3154 ± 0.0082 |
| PAM | 0.3782 ± 0.0038 | 0.3459 ± 0.0128 | 0.3939 ± 0.0096 | - |
| **SSL-6M (ours)** | **0.0148 ± 0.0020** | **0.0757 ± 0.0033** | **0.0550 ± 0.0037** | **0.0489 ± 0.0025** |
| **SSL-120K (ours)** | 0.0311 ± 0.0025 | 0.0878 ± 0.0041 | 0.0678 ± 0.0038 | 0.0600 ± 0.0028 |

| | Document Type ($\alpha = 3$) | | | |
|---|---|---|---|---|
| Method | Pure | LDA | CTM | PAM |
| LDA | 0.0561 ± 0.0030 | - | 0.1942 ± 0.0156 | 0.1813 ± 0.0146 |
| CTM | 0.2347 ± 0.0048 | 0.2299 ± 0.0082 | - | 0.3362 ± 0.0086 |
| PAM | 0.4031 ± 0.0046 | 0.3665 ± 0.0117 | 0.4510 ± 0.0100 | - |
| **SSL-6M (ours)** | **0.0308 ± 0.0070** | **0.0899 ± 0.0053** | **0.0799 ± 0.0048** | **0.0712 ± 0.0038** |
| **SSL-120K (ours)** | 0.0546 ± 0.0097 | 0.1292 ± 0.0067 | 0.1062 ± 0.0058 | 0.0988 ± 0.0048 |

| | Document Type ($\alpha = 5$) | | | |
|---|---|---|---|---|
| Method | Pure | LDA | CTM | PAM |
| LDA | 0.0731 ± 0.0040 | - | 0.2123 ± 0.0159 | 0.1995 ± 0.0144 |
| CTM | 0.2655 ± 0.0059 | 0.2413 ± 0.0090 | - | 0.3428 ± 0.0090 |
| PAM | 0.4337 ± 0.0055 | 0.3608 ± 0.0089 | 0.4794 ± 0.0097 | - |
| **SSL-6M (ours)** | **0.0501 ± 0.0030** | **0.1041 ± 0.0062** | **0.0970 ± 0.0057** | **0.0787 ± 0.0043** |
| **SSL-120K (ours)** | 0.0832 ± 0.0174 | 0.1629 ± 0.0082 | 0.1245 ± 0.0071 | 0.1077 ± 0.0054 |

| | Document Type ($\alpha = 7$) | | | |
|---|---|---|---|---|
| Method | Pure | LDA | CTM | PAM |
| LDA | 0.0883 ± 0.0050 | - | 0.2438 ± 0.0169 | 0.2138 ± 0.0158 |
| CTM | 0.2840 ± 0.0060 | 0.2556 ± 0.0089 | - | 0.3530 ± 0.0099 |
| PAM | 0.4561 ± 0.0052 | 0.3707 ± 0.0089 | 0.4979 ± 0.0101 | - |
| **SSL-6M (ours)** | **0.0391 ± 0.0022** | **0.1233 ± 0.0071** | **0.1071 ± 0.0068** | **0.0960 ± 0.0057** |
| **SSL-120K (ours)** | 0.1219 ± 0.0227 | 0.1851 ± 0.0079 | 0.1340 ± 0.0078 | 0.1358 ± 0.0070 |

| | Document Type ($\alpha = 9$) | | | |
|---|---|---|---|---|
| Method | Pure | LDA | CTM | PAM |
| LDA | 0.1051 ± 0.0065 | - | 0.2559 ± 0.0162 | 0.2281 ± 0.0152 |
| CTM | 0.3129 ± 0.0079 | 0.2580 ± 0.0092 | - | 0.3556 ± 0.0084 |
| PAM | 0.4805 ± 0.0071 | 0.3776 ± 0.0093 | 0.5156 ± 0.0088 | - |
| **SSL-6M (ours)** | **0.0517 ± 0.0045** | **0.1358 ± 0.0089** | **0.1101 ± 0.0064** | **0.0971 ± 0.0053** |
| **SSL-120K (ours)** | 0.1121 ± 0.0202 | 0.2221 ± 0.0123 | 0.1449 ± 0.0077 | 0.1460 ± 0.0070 |

Table C.1: TV between our recovered topic posterior and the true topic posterior of our self-supervised learning approach versus topic inference via Markov Chain Monte Carlo assuming a specific prior for $\alpha = 1, 3, 5, 7, 9$. The 95% confidence interval is reported.

Table C.1 shows that our approach with 6 million training documents consistently outperforms mis-specified topic priors by yielding a much lower TV distance from the true topic posterior. Note that the entries with correct topic model are omitted, because in our experiment the Markov Chain Monte Carlo recovered topic posterior assuming the correct topic model is exactly what we use as

our ground truth topic posterior. Meanwhile, our approach with 120 thousand training documents outperforms misspecified prior in almost every scenario, except when compared to LDA prior on pure topic documents. This is is likely because the LDA topic prior is relatively concentrated on one specific topic and thus similar to the pure topic prior.

We also present a comparison between our self-supervised learning approach and posterior inference assuming a specific topic prior on the task of recovering major topics in the topic proportion for $\alpha = 1, 3, 5, 7, 9$, as a supplement to Section 5.3. As shown in Table C.2, in almost every scenario we test on, our model can perform competitively against the MCMC-inferred posterior assuming the correct topic prior and outperform misspecified topic prior.

| Method | Document Type ($\alpha = 1$) | | |
| --- | --- | --- | --- |
| | LDA | CTM | PAM |
| LDA | **0.9150 ± 0.0387** | 0.8850 ± 0.0344 | 0.8500 ± 0.0360 |
| CTM | 0.9000 ± 0.0416 | **0.9175 ± 0.0275** | 0.8300 ± 0.0370 |
| PAM | 0.8750 ± 0.0458 | 0.7250 ± 0.0397 | 0.9050 ± 0.0320 |
| **SSL-6M (ours)** | 0.9050 ± 0.0406 | **0.9175 ± 0.0275** | 0.9025 ± 0.0330 |
| **SSL-120K (ours)** | **0.9150 ± 0.0387** | 0.9100 ± 0.0292 | **0.9200 ± 0.0289** |

| Method | Document Type ($\alpha = 3$) | | |
| --- | --- | --- | --- |
| | LDA | CTM | PAM |
| LDA | **0.8900 ± 0.0434** | 0.8200 ± 0.0354 | 0.8200 ± 0.0404 |
| CTM | 0.8800 ± 0.0450 | **0.8900 ± 0.0326** | 0.7650 ± 0.0404 |
| PAM | 0.8550 ± 0.0488 | 0.6375 ± 0.0392 | **0.8950 ± 0.0357** |
| **SSL-6M (ours)** | 0.8750 ± 0.0458 | **0.8900 ± 0.0311** | **0.8950 ± 0.0344** |
| **SSL-120K (ours)** | 0.8700 ± 0.0466 | **0.8900 ± 0.0312** | **0.8950 ± 0.0357** |

| Method | Document Type ($\alpha = 5$) | | |
| --- | --- | --- | --- |
| | LDA | CTM | PAM |
| LDA | **0.9050 ± 0.0406** | 0.8500 ± 0.0339 | 0.7750 ± 0.0390 |
| CTM | **0.9050 ± 0.0406** | **0.8975 ± 0.0304** | 0.6775 ± 0.0421 |
| PAM | 0.8850 ± 0.0442 | 0.6150 ± 0.0371 | **0.8825 ± 0.0379** |
| **SSL-6M (ours)** | 0.9000 ± 0.0416 | **0.8975 ± 0.0296** | 0.8675 ± 0.0407 |
| **SSL-120K (ours)** | 0.8800 ± 0.0450 | 0.8900 ± 0.0303 | 0.8750 ± 0.0390 |

| Method | Document Type ($\alpha = 7$) | | |
| --- | --- | --- | --- |
| | LDA | CTM | PAM |
| LDA | 0.8800 ± 0.0450 | 0.7750 ± 0.0397 | 0.7475 ± 0.0433 |
| CTM | 0.8650 ± 0.0474 | **0.8825 ± 0.0353** | 0.6250 ± 0.0390 |
| PAM | 0.8450 ± 0.0502 | 0.5725 ± 0.0356 | 0.8700 ± 0.0411 |
| **SSL-6M (ours)** | 0.9150 ± 0.0387 | 0.8675 ± 0.0383 | 0.8675 ± 0.0407 |
| **SSL-120K (ours)** | **0.9350 ± 0.0342** | 0.8775 ± 0.0370 | **0.8725 ± 0.0398** |

| Method | Document Type ($\alpha = 9$) | | |
| --- | --- | --- | --- |
| | LDA | CTM | PAM |
| LDA | 0.8850 ± 0.0442 | 0.7750 ± 0.0384 | 0.7350 ± 0.0432 |
| CTM | 0.8950 ± 0.0425 | 0.8800 ± 0.0348 | 0.5925 ± 0.0392 |
| PAM | 0.8650 ± 0.0474 | 0.5675 ± 0.0358 | **0.8600 ± 0.0428** |
| **SSL-6M (ours)** | **0.9100 ± 0.0397** | **0.8850 ± 0.0330** | 0.8325 ± 0.0461 |
| **SSL-120K (ours)** | 0.8600 ± 0.0481 | 0.8800 ± 0.0341 | 0.8425 ± 0.0430 |

Table C.2: Major topic recovery rate of our approach versus posterior inference via Markov Chain Monte Carlo assuming a specific prior for $\alpha = 1, 3, 5, 7, 9$. We report the 95% confidence interval.

To further investigate the performance of self-supervised learning, we compare our approach with Variational Inference assuming a specific topic prior on the task of recovering major topics in test document's topic proportion. Table C.3 shows that, similar to our findings from Table C.2, our model can perform competitively against the Variational Inference posterior assuming the correct

topic prior, particularly for relative small $\alpha$ values, and can outperform misspecified topic prior in almost every test scenario.

| Method | Document Type ($\alpha = 1$) | | |
|---|---|---|---|
| | LDA | CTM | PAM |
| LDA | $0.8950 \pm 0.0424$ | $0.8325 \pm 0.0382$ | $0.8625 \pm 0.0339$ |
| CTM | $0.9050 \pm 0.0410$ | $0.9125 \pm 0.0283$ | $0.8150 \pm 0.0354$ |
| PAM | $0.8600 \pm 0.0481$ | $0.7125 \pm 0.0382$ | $0.9050 \pm 0.0325$ |
| **SSL-6M (ours)** | $0.9050 \pm 0.0406$ | **$0.9175 \pm 0.0275$** | $0.9025 \pm 0.0330$ |
| **SSL-120K (ours)** | **$0.9150 \pm 0.0387$** | $0.9100 \pm 0.0292$ | **$0.9200 \pm 0.0289$** |

| Method | Document Type ($\alpha = 3$) | | |
|---|---|---|---|
| | LDA | CTM | PAM |
| LDA | **$0.8750 \pm 0.0452$** | $0.7325 \pm 0.0396$ | $0.7875 \pm 0.0382$ |
| CTM | $0.8800 \pm 0.0450$ | $0.8800 \pm 0.0325$ | $0.6975 \pm 0.0410$ |
| PAM | $0.8550 \pm 0.0488$ | $0.6075 \pm 0.0354$ | $0.8825 \pm 0.0367$ |
| **SSL-6M (ours)** | **$0.8750 \pm 0.0458$** | **$0.8900 \pm 0.0311$** | **$0.8950 \pm 0.0344$** |
| **SSL-120K (ours)** | $0.8700 \pm 0.0466$ | **$0.8900 \pm 0.0312$** | **$0.8950 \pm 0.0357$** |

| Method | Document Type ($\alpha = 5$) | | |
|---|---|---|---|
| | LDA | CTM | PAM |
| LDA | $0.8800 \pm 0.0452$ | $0.7575 \pm 0.0354$ | $0.7375 \pm 0.0410$ |
| CTM | $0.8750 \pm 0.0452$ | **$0.8975 \pm 0.0283$** | $0.5925 \pm 0.0396$ |
| PAM | $0.8500 \pm 0.0495$ | $0.5675 \pm 0.0297$ | **$0.8875 \pm 0.0368$** |
| **SSL-6M (ours)** | **$0.9000 \pm 0.0416$** | **$0.8975 \pm 0.0296$** | $0.8675 \pm 0.0407$ |
| **SSL-120K (ours)** | $0.8800 \pm 0.0450$ | $0.8900 \pm 0.0303$ | $0.8750 \pm 0.0390$ |

| Method | Document Type ($\alpha = 7$) | | |
|---|---|---|---|
| | LDA | CTM | PAM |
| LDA | $0.9250 \pm 0.0368$ | $0.7175 \pm 0.0396$ | $0.6875 \pm 0.0410$ |
| CTM | $0.9300 \pm 0.0354$ | $0.8700 \pm 0.0382$ | $0.5700 \pm 0.0354$ |
| PAM | $0.9050 \pm 0.0410$ | $0.5400 \pm 0.0325$ | **$0.8750 \pm 0.0396$** |
| **SSL-6M (ours)** | $0.9150 \pm 0.0387$ | $0.8675 \pm 0.0383$ | $0.8675 \pm 0.0407$ |
| **SSL-120K (ours)** | **$0.9350 \pm 0.0342$** | **$0.8775 \pm 0.0370$** | $0.8725 \pm 0.0398$ |

| Method | Document Type ($\alpha = 9$) | | |
|---|---|---|---|
| | LDA | CTM | PAM |
| LDA | $0.9050 \pm 0.0410$ | $0.7250 \pm 0.0396$ | $0.6975 \pm 0.0410$ |
| CTM | **$0.9200 \pm 0.0382$** | $0.8725 \pm 0.0339$ | $0.5425 \pm 0.0368$ |
| PAM | $0.8750 \pm 0.0452$ | $0.5300 \pm 0.0325$ | **$0.8425 \pm 0.0452$** |
| **SSL-6M (ours)** | $0.9100 \pm 0.0397$ | **$0.8850 \pm 0.0330$** | $0.8325 \pm 0.0461$ |
| **SSL-120K (ours)** | $0.8600 \pm 0.0481$ | $0.8800 \pm 0.0341$ | **$0.8425 \pm 0.0430$** |

Table C.3: Major topic recovery rate of our approach versus posterior inference via Variational Inference assuming a specific prior for $\alpha = 1, 3, 5, 7, 9$. We report the 95% confidence interval.

## C.4 THE EFFECT OF TRAINING EPOCHS

We measure how the number of training epochs may influence topic posterior recovery loss by varying the number of training epochs. The results we present in Section 5 are based on models trained for 200 epochs for all types of documents. Here, we present the topic recovery loss, measured as the Total Variation distance between our recovered topic posterior and the true topic posterior, for epochs=25, 50, 100, 200 using the same model architecture and same sampling scheme as in Section 5.

Table C.4 shows that topic posterior recovery loss steadily decreases as the number of epochs gets larger. Interestingly, even with just training 50 epochs, our model can recover the topic posterior

within a Total Variation distance of less than 0.3 off the true topic posterior for documents generated from all four topic models.

| Document type | Number of epochs | Dirichlet hyperparameter $\alpha$ | | | | |
|---|---|---|---|---|---|---|
| | | 1 | 3 | 5 | 7 | 9 |
| Pure-topic | 25 | 0.0505 | 0.1025 | 0.1756 | 0.6222 | 0.8285 |
| | 50 | 0.0310 | 0.0744 | 0.0991 | 0.1522 | 0.2873 |
| | 100 | 0.0160 | 0.0363 | 0.0511 | 0.0911 | 0.1257 |
| | 200 | 0.0148 | 0.0308 | 0.0501 | 0.0391 | 0.0517 |
| LDA | 25 | 0.1236 | 0.1583 | 0.1812 | 0.2022 | 0.2218 |
| | 50 | 0.0967 | 0.1428 | 0.1387 | 0.1953 | 0.2117 |
| | 100 | 0.0805 | 0.1018 | 0.1101 | 0.1361 | 0.1538 |
| | 200 | 0.0709 | 0.0951 | 0.1045 | 0.1222 | 0.1377 |
| CTM | 25 | 0.1089 | 0.1599 | 0.1563 | 0.1747 | 0.1971 |
| | 50 | 0.0900 | 0.1132 | 0.1425 | 0.1455 | 0.1501 |
| | 100 | 0.0623 | 0.0948 | 0.1102 | 0.1187 | 0.1243 |
| | 200 | 0.0550 | 0.0799 | 0.0970 | 0.1071 | 0.1101 |
| PAM | 25 | 0.1072 | 0.1342 | 0.1480 | 0.1688 | 0.1754 |
| | 50 | 0.0820 | 0.1036 | 0.1246 | 0.1185 | 0.1395 |
| | 100 | 0.0543 | 0.0755 | 0.0859 | 0.1087 | 0.1095 |
| | 200 | 0.0436 | 0.0659 | 0.0787 | 0.0953 | 0.0971 |

Table C.4: Topic posterior recovery loss of our self-supervised learning approach, measured in Total Variation distance, for all four types of documents and $\alpha = 1, 3, 5, 7, 9$. The number of training epochs ranges from 25 to 200, and we sample 60K new training documents in every 2 epochs, which corresponds to 720K to 6M training documents.

## C.5 HYPERPARAMETER TUNING EXPERIMENTS

As described in Section 5, the two types of neural network architecture we use are fully-connected neural networks and attention-based neural networks (see Figure C.3). In this section, we present some of our TV results of different combinations of hyperparameters for both types of models. Note that our attention-based neural network slightly differs from typical transformers (Vaswani et al., 2017) in that we use batch normalization instead of layer normalization in every residual connection. We only use one attention head in each multi-head attention layer. During training, we use AMSGrad optimizer and an initial learning rate of 0.0001 or 0.0002. We reduce the learning rate by 50% whenever validation loss does not decrease in 10 epochs.

The main hyperparameters in our fully-connected neural networks include the hidden dimension, the number of layers, and whether we apply residual connections. Table C.5 shows the TV distance results when we vary these hyperparameters for the LDA scenario when $\alpha = 1$. However, fully-connected neural networks perform poorly in CTM and PAM case. For instance, when $\alpha = 1$ and hidden dimension = 4096, we find that hyperparameters that work well in the LDA case no longer yield satisfactory results (Table C.6).

| TV | Hidden dimension | | |
|---|---|---|---|
| # layers (residual) | 1024 | 2048 | 4096 |
| 3 | 0.1509 | 0.1112 | 0.1095 |
| 6 | 0.7902 | 0.7893 | 0.7901 |
| 3 (residual) | 0.0871 | 0.0867 | 0.0862 |
| 6 (residual) | 0.0822 | 0.0835 | **0.0709** |

Table C.5: Fully-connected neural network's performance in the LDA $\alpha = 1$ scenario.

For attention-based neural networks applied to CTM and PAM scenarios, we mainly consider the number of layers and the hidden dimension per attention layer. Table C.7 presents the TV distance between recovered topic posterior and true topic posterior in the CTM setting, with varying number of layers and attention dimension for $\alpha = 1, 3, 5$ and using a more coarse estimate to the true topic posterior than what we use as our final true CTM topic posterior. We find that the model performs

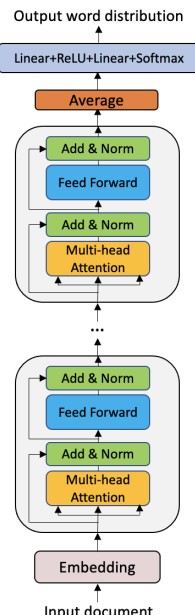

Figure C.3: Our attention-based neural network architecture

| TV | Layer Type | |
|---|---|---|
| # layers | regular | residual |
| 3 | 0.1824 | 0.1665 |
| 4 | 0.1724 | 0.1656 |
| 6 | 0.1700 | 0.1556 |

Table C.6: Fully-connected neural network's performance in the CTM $\alpha = 1$ scenario with 4096 hidden dimensions.

the best when attention dimension is 768 or 1024, and when the attention dimension increases to 2048 the model gives a higher TV (see Table C.8).

| TV | | # layers | | |
|---|---|---|---|---|
| $\alpha$ | Attention dimension | 4 | 6 | 8 |
| 1 | 768 | 0.0902 | 0.0844 | **0.0814** |
| | 1024 | 0.0851 | 0.0890 | 0.0836 |
| 3 | 768 | 0.1471 | 0.1429 | 0.1398 |
| | 1024 | 0.1440 | 0.1384 | **0.1383** |
| 5 | 768 | 0.1794 | 0.1767 | **0.1708** |
| | 1024 | 0.1878 | 0.1787 | 0.1782 |

Table C.7: Attention-based neural network's performance on CTM documents for $\alpha = 1, 3, 5$.

| TV | Attention dimension | | |
|---|---|---|---|
| $\alpha$ | 768 | 1024 | 2048 |
| 3 | 0.1471 | **0.1440** | 0.1670 |
| 5 | **0.1794** | 0.1878 | 0.1922 |

Table C.8: 4-layer attention-base neural network's performance on CTM documents when attention layer's dimension varies, for $\alpha = 3, 5$.

## C.6 TOPIC POSTERIOR RECOVERY LOSS AND CONDITION NUMBER

We have proved in Theorem 4 that the upper bound of topic posterior recovery loss depends on the $\ell_1$ Condition Number $\kappa(A^\dagger)$. In this section, we show that $\kappa(A^\dagger)$ is small for every topic matrix $A$ in our experiments (Figure C.4). Note that we use the same topic matrix $A$ for pure-topic model and the LDA model. For CTM and PAM, we use the same topic matrix up to reordering of the topics that corresponds with pairwise topic correlations. Therefore, pure-topic model and LDA model share the same $\kappa(A^\dagger)$, and CTM and PAM share the same $\kappa(A^\dagger)$.

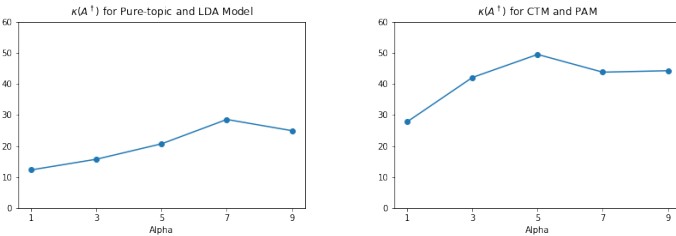

Figure C.4: $\ell_1$ Condition Number for topic matrix $A$

## C.7 VISUALIZING TOPIC CORRELATIONS BETWEEN MODELS

For documents generated by the PAM model, we plotted the estimated posteriors by the self-supervised approach and posterior inference assuming different priors in Figure C.5. From this figure (especially in Documents 3 and 4), one can qualitatively see that the estimated posterior and the PAM MCMC posteriors are aware of the correlation between topics, while the LDA MCMC posterior fails to take the correlation into consideration and hence is more different from the ground truth topic proportions.

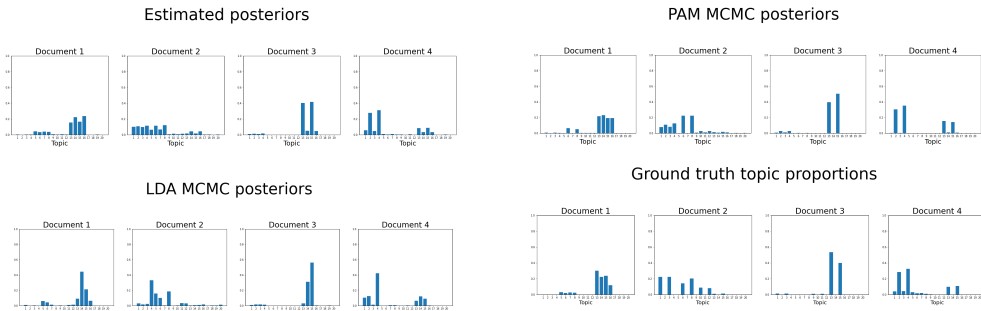

Figure C.5: Comparison of estimated, PAM MCMC, LDA MCMC, and ground truth posterior topic proportions for sample test documents generated by PAM. Overall, across our set of 200 test documents, the estimated posteriors were able to outperform the mis-specified LDA posteriors.

# D    REAL-DATA EXPERIMENT ADDITIONAL DETAILS

In this section, we give more details about our real data experiments in Section 6. In Section D.1, we describe the data processing. We give a more detailed description of baselines and how we extracted our representations in Section D.2. In Section D.3, we report more results using different model architectures.

## D.1    DATA PROCESSING

Here we detailed our usage of the AG news dataset Zhang et al. (2015). Each category has 30,000 samples in the training set and 19,000 samples in the testing set. We first preprocessed the data by removing punctuation and words that occurred in fewer than 10 documents, obtaining in a vocabulary of around 16,700 words, in a similar fashion done by Tosh et al. (2020).

To split the data set into unsupervised dataset and supervised dataset. For each category of documents, we selected a random sample of 1000 documents as labeled supervised dataset while the remaining 116,000 documents fall into unsupervised dataset for representation learning.

## D.2    EXTRACTING REPRESENTATIONS

To give some details about two baseline representations we used, we described in details as follows:

- **Bag of Words (BOW):** For a single document, we constructs BOW embedding by creating a bag-of-words frequency vector of the dimension of vocabulary size, where each entry $i$ represent the frequency of words with id $i$ in that particular document.
- **Word2vec:** To generate word2vec representation fitted the Skip-gram word embedding model on unsupervised dataset (Mikolov et al. (2013)). The implementation is done through the Gensim library (Rehurek & Sojka (2011)), where we used an embedding dimension of 300 and window size of 5. Representation is taken as the average of all trained word embeddings in a single document.

For our own self-supervised method, to extract a representation, we attempted at 1) *softmax+last layer*: apply a Softmax directly to the last layer output, 2) *word2vec+last layer*: apply an word2vec embedding matrix to last layer output after Softmax to reduce dimension (the word2vec matrix is trained over the unsupervised dataset), and 3) *softmax+second2last layer*: apply a Softmax function on top of the second-to-last layer (equivalent to applying an identity matrix to replace the last layer). The dimension of representation with word2vec is 300 while the softmax + second2 last layer has a dimension of 4096. The original last layer representation has a dimension of the vocabulary size (around 16,700), and has shown inferior results than those with dimension reduction techniques.

We included a comparison of the two approaches with dimension reduction in Table D.1, from which we observe that a Softmax on the second-to-last layer output performed better across layers of 3,4 and 5. We reported the result using the *softmax+second2last layer* in Figure 2.

| Test Accuracy | Method | |
| --- | --- | --- |
| # layers (residual) | *word2vec+last layer* | *softmax+second2last layer* |
| 3 | 0.8508 | **0.8714** |
| 4 | 0.8450 | **0.8621** |
| 5 | 0.8492 | **0.8689** |

Table D.1: Test Accuracy for different representation extraction method. We fixed the rest of hyperparameters to be: 5000 embedding dimension, 150 epochs, 0.0002 learning rate, sampling 4 words in labels, weight decay of 0.01 and resample rate of 2.

## D.3    MODEL ARCHITECTURE

We include here more details on our model architecture used in real data experiments. We used residual model, and tested two main hyperparameters: number of residual blocks and hidden dimension size. We fixed the rest of hyperparameters: we used a 5000 embedding dimension and trained

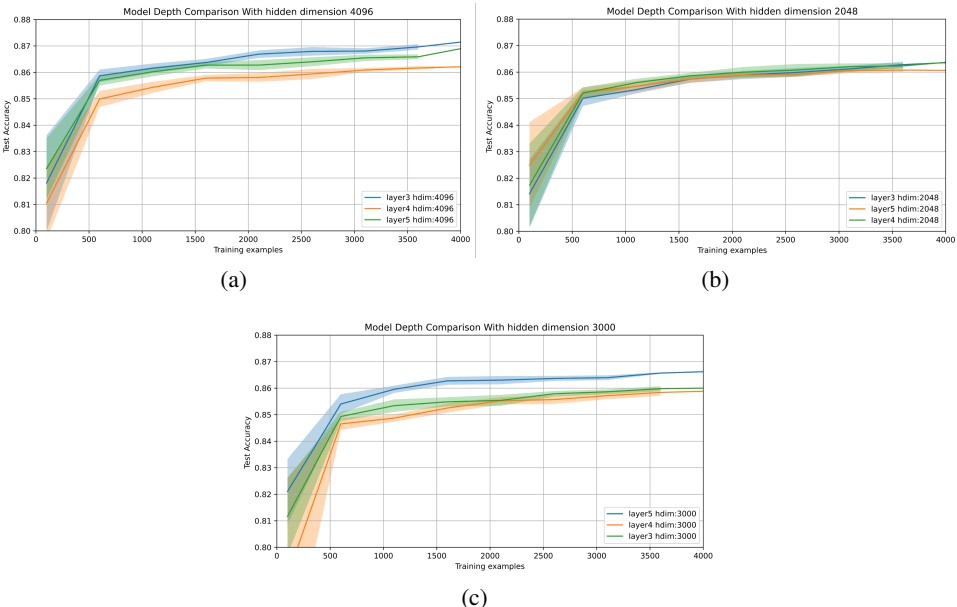

Figure D.1: RBL representation performance varies with different residual model capacity: the best performing is the one with 3 layers and 4096 hidden dimensions, and it was what we used for Figure 2. In all these runs, we use Softmax on second-to-last layer to obtain our representations

for 150 epochs, with 0.0002 learning rate, weight decay of 0.01 and resample rate of 2. We sampled 4 words in labels for variance reduction purpose.

**Varying Depth and Width** We investigated both the effect of model depth and width on the performance of RBL representation. We trained networks of width 2048, 3000, and 4096 nodes respectively, where the number of node refers to the number of node in the linear layer inside a residual block. We varied the depth of the network by choosing to place 3, 4 and 5 of such block.

As shown in Figure D.1, it appears that using wider models in the unsupervised phase leads to better performance when training a linear classifier on the learned representations. Number of residual block does not seem to have a clear relationship with test accuracy from limited experiments we ran. The best accuracy is achieved when we have 3 residual blocks with dimension of 4096, which is what we used to generate results in Figure 2.

