# OpenReview forum: "One Objective for All Models --- Self-supervised Learning for Topic Models"
_ICLR.cc/2022/Conference — ICLR 2022 Submitted_

### Official Review · Reviewer_b3P7 · 2021-10-28

**Correctness:** 3
**Technical Novelty And Significance:** 3
**Empirical Novelty And Significance:** 2
**Recommendation:** 6
**Confidence:** 3

**Main Review:**

- The main strength of this paper lies in the thorough theoretical analysis and corresponding proofs. The authors not only prove that a new reconstruction-based objective can also extract posterior topic information for a general topic model, but also strengthen the guarantee for contrastive objective in Tosh et al. (2020) by removing some of their assumptions and the necessity of landmark documents.

- While the theoretical analysis is interesting, objective and motivation of this paper is not entirely clear and convincing. It seems there is lack of outlook for future work where this analysis could be useful, and the empirical study on real data is also somewhat weak.  There is no comparison with more recent self-supervised and contextual document representation work, such as BERT mentioned in the beginning of the paper. Actually this is also a common issue in both Arora et al. (2019) and Tosh et al. (2000), which are two most closely related papers.
    - The LDA “is actually doing very poor on several ‘objectively’ evaluable predictive tasks”, because “it is not designed, nor trained for such tasks, such as classification, there is not warrantee that the estimated topic vector is good at discriminating documents" (http://www.cs.cmu.edu/~epxing/talks/ACL2.pdf).
    - BOW, word2vec and NCE are weak base lines far behind more recent self-supervised and contextual representation models for document classification.
    - The experiments are not conducted on a fair basis. For example, word2vec is a light-weighted neural network with only one layer, but this paper implements the proposed model using some deep neural networks including 8 heavy-weighted Transformer blocks, which still demonstrate no significant difference.

- typos
    - "constrastive" in Page 1


**Summary Of The Paper:**

The main strength of this paper lies in the thorough theoretical analysis and corresponding proofs. The authors not only prove that a new reconstruction-based objective can also extract posterior topic information for a general topic model, but also strengthen the guarantee for contrastive objective in Tosh et al. (2020) by removing some of their assumptions and the necessity of landmark documents.

**Summary Of The Review:**

The theoretical analysis is interesting, but the motivation of analysis and evaluations are not entirely convincing. I suggest to focus on theoretical analysis and clarify where this analysis could be useful.

---

> ### Author Response · Authors · 2021-11-16
> **Response**
>
>
> We would like to thank the reviewer for the detailed comments. We will try to address the concerns below. We also make several clarifications in a separate post to address some common concerns for all the reviewers.
>
> - Comparison with recent self-supervised and contextual document representation work (e.g., BERT). Common issue in current paper, Arora et al. (2019) and Tosh et al. (2020).
>
>     As we clarified in the post to all reviewers, we do not claim that SSL can achieve state-of-the-art performance. Our focus is to understand why optimizing the SSL objective can extract useful information to solve the downstream task. Whether one optimizes this objective using a huge model (e.g., BERT) or a smaller model like we do in our paper is not the main point of the discussion. In fact, our experiments show that even for a simple model like what we use, as long as it achieves good performance on the SSL objectives, it can extract reasonable information. Of course, a larger model could potentially get much better objective, and our theory does not give answers to how large the model needs to be in order to minimize the SSL objective in practice, we view that as an orthogonal future direction.
>
> - Thanks for pointing out the typo. We will fix it in the revision.

---

> > ### Comment · Reviewer_b3P7 · 2021-11-16
> > **Concern on where this analysis could be useful for any practical (downstream) task**
> >
> > The authors' responses confirm that the main contribution of this paper is the theoretical analysis on connections between traditional topic models and specific self-supervised methods in terms of topic posterior. But the responses didn't answer the main concern on where this analysis could be useful for any practical (downstream) task. Section 6 demonstrated that the proposed method is not practically useful for downstream tasks. Perhaps removing section 6 would make the main point of this paper even clearer.

---

> > > ### Author Response · Authors · 2021-11-27
> > > **Response**
> > >
> > > We appreciate the reviewer for providing the feedback. As we have clarified in the general post, we do not claim that we design a better method than others in practice. Instead, the goal and contribution of current paper is to give a reason why SSL can provide useful information for downstream task in the setting of topic models.

---

### Official Review · Reviewer_HZxy · 2021-11-01

**Correctness:** 3
**Technical Novelty And Significance:** 3
**Empirical Novelty And Significance:** 2
**Recommendation:** 6
**Confidence:** 3

**Main Review:**

* Topic-word weight matrix $A$: It is known that one advantage of topic models is that each latent topic can be represented as a set of words. The topic-word weight matrix is learnt together with the posterior of the topic proportion vector $w$. All the theorems assume that $w$ is a condition, and it seems that $\theta$ is the left pseudo-inverse of $A$. Looking at existing topic models, either probabilistic or neural, it is not clear to me how $A$ is learned by the network models used in the experiments.
* Neural topic modelling: How is this work related to neural topic models based on autoencoding variational inference, for instance, by Srivastava and Sutton 2017? It might be good to see how a neural topic model can recover the topic posterior.
* The synthetic experimental results seem to say that the reconstruction-based objective can recover the posterior of $w$ given a document. What is the implication of this from a topic modelling practitioner's point of view? Can one use the reconstruction-based objective with neural networks in place of topic models? However, it is known that one advantage of topic models, particularly the probabilistic ones, is that they can learn interpretable topics. Besides the approximation of the posterior of $w$, I cannot see how the proposed theorems could benefit the topic modelling community.
* It is an interesting statement that "attention-based architecture performs the best for recovering topic posterior distribution for CTM and PAM documents". I wonder if the authors can discuss this more, like the possible reasons.
* The experiments on the real dataset seems to show that the self-supervised objectives can learn useful representation for document classification. Again, I would like to see how neural topic models perform in this scenario. Furthermore, compared with the other embedding method, like the language models, what is the advantage of this work?

**Summary Of The Paper:**

This paper shows theoretically and empirically that self-supervised objectives can be used to extract useful information about the posterior of the topic proportion vector given a document, regardless of the underlying models. It extends the findings of Tosh et al.2020 about contrastive objectives to reconstruction-based objectives. It is interesting to see that one simple reconstruction objective recover the posteriors generated from the different probabilistic topic models on the synthetic dataset.

**Summary Of The Review:**

Overall, showing that both the reconstruction-based objective and the contrastive object can recover the posterior of the topic proportion vector is interesting. It would be good to demonstrate how this theoretical finds can benefit the wide topic modeling community with more experiments on real-world datasets.

---

> ### Author Response · Authors · 2021-11-16
> **Rsponse**
>
>
> We would like to thank the reviewer for the detailed review. We will attempt to address the concerns below. We also make several clarifications in a separate post to address some common concerns for all the reviewers.
>
> - Topic-word weight matrix $A$.
>
>     In the synthetic experiments, the true value of $A$ is given. Hence, to recover the estimated topic posterior from the learned representation $f(x)$, we can directly use $A^\dagger f(x)$ as the estimation. In the real data experiments, as our goal is not to recover the topic posterior but to solve the downstream task, we do not need to know the value $A$.
>
> - Compare with neural topic modelling and other methods (e.g. language models).
>
>     As we clarified in the post to all reviewers, our focus is not the inference problem nor to show self-supervised learning (SSL) can achieve the state-of-the-art performance. The problem we are interested in is to understand why SSL can extract useful information to the downstream task. Therefore, the focus of current paper is in fact different from the problem that these works/methods want to study.
>
> - Implication to the topic modelling community.
>
>     As we clarified in the post to all reviewers, we do not claim SSL can achieve state-of-the-art performance for the inference problem. Our focus is to show SSL can extract useful information to the downstream task. We believe one benefit that offers from SSL is that it can be oblivious to the specific topic model, and hence is less susceptible to model misspecification. As we shown in the synthetic experiments, traditional inference approach suffer from the problem of misspecification and cannot provide useful information in this case. On the other hand, SSL can perform reasonably well across different models, even when comparing with the inference method with correct prior.
>
> - Potential reasons for attention-based method performs better in CTM and PAM case.
>
>     One potential reason is that the topics are correlated in our setting of CTM and PAM, and attention is able to capture such information. Intuitively, attention is designed to capture the dependency of the words in the sequence. On the other hand, it might be hard for vanilla fully-connected neural networks to capture such information.

---

### Official Review · Reviewer_KozZ · 2021-11-02

**Correctness:** 3
**Technical Novelty And Significance:** 3
**Empirical Novelty And Significance:** 2
**Recommendation:** 5
**Confidence:** 4

**Main Review:**


Strengths
---------
+ Takes on an interesting and timely topic (self-supervised learning for probabilistic models)
+ Broad scope of objectives, looking at both "reconstruction" and "contrastive" learning
+ Broad scope of topic models, considering a "general" family that includes LDA, CTM, and Pachinko Allocation as special cases
+ Seems to extend the "nearest" previous work (Tosh et al. 2020) meaningfully, focusing on reconstruction objectives not just contrastive learning

Weaknesses
----------

I list 5 concerns here, with detailed discussion and questions for the authors below

- W1: While theorems suggest "existence" of a linear transformation that will approximate the posterior, the actual construction procedure for the "recovered topic posterior" is unclear

- W2: Many steps are difficult to understand / replicate from main paper

- W3: Unclear what theorems can say about finite training sets

- W4: Justification / intuition for Theorems is limited in the main paper

Responses to W1-W3 are most important for the rebuttal.


## W1: Actual procedure for constructing the "recovered topic posterior" is unclear

In both synthetic and real experiments, the proposed self-supervised learning (SSL) method is used to produce a "recovered topic posterior" p( w | x). However, the procedure used here is unclear... how do we estimate p( w | x) using the learning function f(x)?

The theorems imply that a linear function *exists* with limited (or zero) approximation error for any chosen scalar summary of the doc-topic weights w. However, how such a linear function is constructed is unclear. The bottom of page four suggests that when t=1 and A is full rank, that "one can use the pseudoinverse of A to recover the posterior", however it seems (1) unclear what the procedure is in general and what its assumptions are, and (2) odd that the prior may not needed at all.

*Can the authors clarify how to estimate the recovered topic posterior using the proposed SSL method?*


## W2: Many other steps are difficult to understand / replicate from main paper

Here's a quick list of questions on experimental steps I am confused about / would have trouble reproducing

For the toy experiments in Sec. 5:

- Do you estimate the topic-word parameter A? Or assume the true value is given?
- What is the format for document x provided as input to the neural networks that define f(x)? The top paragraph of page 7 makes it seem like you provide an ordered list of words. Wouldn't a bag-of-words count vector be a more robust choice?
- How do you set t=1 (predict one word given others) but somehow also use "the last 6 words are chosen as the prediction target"?
- How do you estimate the "recovered topic posterior" for each individual model (LDA, CTM, etc)? Is this also using HMC (which is used to infer the ground-truth posterior)?
- Why use 2000 documents for the "pure" topic model but 500 in test set for other models? Wouldn't more complex models benefit from a larger test set?

For the real experiments in Sec. 6:

- How many topics were used?
- How did you get topic-word parameters for this "real" dataset?
- How big is the AG news dataset? Main paper should at least describe how many documents in train/test, and how many vocabulary words.


## W3:  Unclear what theorems / methods can say about finite training sets

All the theorems seem to hold when considering terms that are expectations over a known distribution over observed-data x and missing-data y. However, in practical data analysis we do not know the true data generating distribution, we only have a finite training set.

I am wondering about this method's potential in practice for modest-size datasets. For the synthetic dataset with V=5000 (a modest vocabulary size), the experiments considered 0.72 million to 6 million documents, which seems quite large.

*What practically must be true of the observed dataset for the presented methods to work well?*


## W4: Justification / intuition for Theorems is limited in the main paper

All 3 theorems in the main paper are presented without much intuition or justification about why they should be true, which I think limits their impact on the reader. (I'll try to wade thru the supplement, but did not have time before the review deadline).

Theorem 3 tries to give intuition for the t=1 case, but I think could be stronger: why should f(x) have an optimal form $p( y=v_1 | x)$? Why should "these probabilities" have the form $A E[ w | x]$? I know space is limited, but helping your reader figure things out a bit more explicitly will increase the impact.

Furthermore, the reader would benefit from understanding how tight the bounds in Theorem 4 are. Can we compute the bound quality for toy data and understand it more practically?


Detailed Feedback on Presentation
---------------------------------

No need to reply to these in rebuttal but please do address as you see fit in any revision

Page 3:
- "many topic models can be viewed"... should probably say "the generative process of many topic models can be viewed..."

- the definition of A_ij is not quite right. I would not say "word i \in topic j", I would say "word i | topic j". A word is not contained in a topic, Each word has a chance of being generated.

- I'd really avoid writing $\Delta(K)$ and would just use $\Delta$ throughout .... unclear why this needs to be a function of $K$ but the topic-word parameters (whose size also depends on $K$) does not

- Should we call the reconstruction objective a "partial reconstruction" or "masked reconstruction"? I'm used to reconstruction in an auto-encoder context, where the usual "reconstruction" objective is literally to recover all observed data, not a piece of observed data that we are pretending not to see

- In Eq. 1, are you assuming an ordered or unordered representation of the words in x and y?

Page 4:

- I would not reuse the variable y in both reconstruction and contrastive contexts. Find another variable. Same with theta.

Page 5:

- I would use $f^*$ to denote the exact minimizer, not just $f$


Figure 2 caption should clarify:

- what is the takeaway for this figure? Does reader want to see low values? Does this figure suggest the approach is working as expected?
- what procedure is used for the "recovered" posterior? Your proposed SSL method?
- why does Pure have a non-monotonic trend as alpha gets larger?



**Summary Of The Paper:**

This paper considers the problem of performing posterior inference for probabilistic topic models. A "general" topic model consists of two key parts:

- $A$ : the topic-word probability matrix
- $\Delta$ : the prior distribution over doc-topic probability vector w

Given these parts and a new document x, the goal is to infer the posterior over w: $p(w | x)$.

It is well-known that this posterior can be estimated in several ways using standard approximate Bayesian inference methods: MCMC, variational, etc. The detailed steps of these methods are usually customized to the choice of prior.

This paper suggests a new possibility: that a learned transformation -- denoted f(x) -- trained using a so-called *reconstruction* loss (see Eq 1), can be then translated into the desired posterior via a linear function. Surprisingly, the paper suggests that learning f(x) does not depend on the prior \Delta, so the big idea is that this representation is robust to misspecification of the prior.

Concretely, the contributions of the paper seem to be:

1) Theorem 3, which claims that there exist linear weights \theta that, when applied to the vector produced by the ideal learned transformation f(x) that minimizes the reconstruction objective (Eq 1), can exactly equal the expected value of any polynomial summary function of the doc-topic probability vector w under the posterior.

2) Theorem 4, which claims that even if f is not an exact minimizer, if it is within an additive tolerance \epsilon of the ideal loss, then there exist again weights \theta such that the squared error between the true posterior summary and the linear function has bounded error for all documents.

3) Theorem 5, which looks at the contrastive objective in Eq 2, and shows that there exists a linear function that can recover the expected polynomical summaries of random variable w under the posterior.

Experiments in Sec. 5 look at toy data, with results in Fig 2 and Table 1. Essentially, these results show that the proposed SSL method produces lower error posterior estimates (in terms of  total-variation distance from ground truth) than using Bayesian inference with the wrong prior.

Experiments in Sec. 6 look at real data (the AG news dataset), comparing the representations learned by SSL at a downstream classification task to baselines of bag-of-words, word2vec, and the contrastive method of Tosh et al. Fig. 3 suggests that the proposed method offers a few percentage points absolute gain in accuracy over word2vec, and is similar to the previous NCE method (Tosh et al) in accuracy.

**Summary Of The Review:**

Overall I think the direction of the paper is interesting, but the main paper at present is missing some key pieces (esp. how the recovered topic posterior is produced once the SSL representation f is learned). In addition, I have concerns about the disconnect in how theorems might apply to finite training sets in practice, and how little intuition was provided in the paper for the theory.

I could be persuaded to change my mind by a strong rebuttal.

---

> ### Author Response · Authors · 2021-11-16
> **Response**
>
> We would like to thank the reviewer for the detailed feedback. We will attempt to address the concerns below. We also make several clarifications in a separate post to address some common concerns for all the reviewers.
>
> - W1: Actual procedure for constructing the ``recovered topic posterior'' is unclear.
>
>     For synthetic experiments, we would like to first clarify that we do not construct $p(w|x)$ based on $f(x)$, but construct $E[w|x]=\int_w wp(w|x)d w$ based on $f(x)$. Here $E[w|x]$ is in fact the posterior mean of the topic proportion and we use this as our estimation of the true topic proportion. The way of constructing $E[w|x]$ based on $f(x)$ for $t=1$ case follows from the paragraph at the end of page 4, which is simply $E[w|x]=A^\dagger f(x)$. For the procedure of the general $t$ case, it is implied by the proof of Theorem 3 in Section A.2 (especially the second to last paragraph of the proof), which is $vec(W_{post})=\mathcal{A}^\dagger f(x)$. Here $vec(W_{post})$ is the vectorization of the topic posterior tensor $E[w^{\otimes t}|x]$ and $\mathcal{A}=\underbrace{A\otimes A\otimes\cdots\otimes A}_{t \text{ times}}$. For the assumption that make sure the above construction holds, as we specified in the statement of our theorem, we assume the topic-word matrix $A$ is full rank. We do not know whether our result would still be true when $A$ is not full rank. For the topic-word matrices $A$ that are used in experiments, they are full rank matrices and thus we can use the above construction.
>
>     For real data experiments, we do not construct the ``recovered topic posterior'', since our focus is to show SSL can extract useful representation which can be later used for the downstream task. So instead of recovering the topic posterior, we directly used the representation given by self-supervised learning (SSL) to solve the downstream task.
>
> - W2: Many other steps are difficult to understand / replicate from main paper.
>
>     Section 5:
>
>     - Do you estimate the topic-word parameter A? Or assume the true value is given?
>
>         We do not estimate $A$. The true value of $A$ is given in this section.
>
>     - What is the format for document x provided as input to the neural networks that define f(x)? The top paragraph of page 7 makes it seem like you provide an ordered list of words. Wouldn't a bag-of-words count vector be a more robust choice?
>
>         For pure-topic and LDA case, we do use a bag-of-words count as the input vector. However, for the CTM and PAM case, since we are using the attention-based model, it require the input be a sequence instead of a bag-of-words count vector. Therefore, when using the attention-based models, the input is a list of words in random order (as in topic models, order of words does not matter).
>
>     - How do you set t=1 (predict one word given others) but somehow also use "the last 6 words are chosen as the prediction target"?
>
>         We meant to say that for each input sequence $x$, we have 6 training samples $(x,y_1),...,(x,y_6)$. So the original unmasked sequence is $xy_1...y_6$. Note that this is different from training on samples $(x,y_1...y_6)$, which is the $t=6$ case that asks the neural network to predict 6 words together.
>
>     - How do you estimate the "recovered topic posterior" for each individual model (LDA, CTM, etc)? Is this also using HMC (which is used to infer the ground-truth posterior)?
>
>         Yes, for the misspecified models, we estimate the ``recovered topic posterior'' $E[w|x]$ using the MCMC. This is the same method for estimating the ground-truth posterior.
>
>     - Why use 2000 documents for the "pure" topic model but 500 in test set for other models? Wouldn't more complex models benefit from a larger test set?
>
>         The reason is because for the complex models, running MCMC to obtain the ground-truth will take much longer time than the simple topic models. We will give the confidence interval for the results to show that such small test set is sufficient for the claims that we are making in our revision.

---

> > ### Author Response · Authors · 2021-11-16
> > **Response - continue**
> >
> > - W2, Section 6:
> >
> >     - How many topics were used?
> >
> >         As we mention in the first paragraph of Section 6.1, the dataset has 4 topics: world, sports, business, and sci/tech.
> >
> >     - How did you get topic-word parameters for this "real" dataset?
> >
> >         We do not need topic-word matrix $A$ in real data experiments. As we clarified in the post to all reviewers, the focus of current paper is not the inference problem, but to show SSL can learning useful representation that would help to solve the downstream task. Therefore, we do not need to recover the topic posterior, and we do not need $A$ for the real data experiment. Instead, we directly use the representation $f(x)$ to solve the downstream task.
> >
> >     - How big is the AG news dataset? Main paper should at least describe how many documents in train/test, and how many vocabulary words.
> >
> >         Due to the space limit, we defer the details in Section D.1, which is copied below.
> >
> >         ''Here we detailed our usage of the AG news dataset Zhang et al. (2015). Each category has 30,000 samples in the training set and 19,000 samples in the testing set. We first preprocessed the data by removing punctuation and words that occurred in fewer than 10 documents, obtaining in a vocabulary of around 16,700 words, in a similar fashion done by Tosh et al. (2020).
> >
> >         To split the data set into unsupervised dataset and supervised dataset. For each category of documents, we selected a random sample of 1000 documents as labeled supervised dataset while the remaining 116,000 documents fall into unsupervised dataset for representation learning.''
> >
> > - W3: Unclear what theorems / methods can say about finite training sets.
> >
> >     As we clarified in the post to all reviewers, we do not claim that SSL can achieve state-of-the-art performance. Our goal is to understand why SSL can provide useful information to the downstream task. Therefore, all of our theoretical results focus on the population loss, since it is easy to analyze.
> >
> >     While all of our theoretical results are in the setting of population loss, Theorem 4 (robust version of the main result) gives an intuitive explanation in the finite training sets. Intuitively, it suggests that even if we only have an approximate minimizer whose loss is $\epsilon$, then we can still guarantee that the representation $f(x)$ could give a $O(\epsilon)$-close approximation of the target function. In general, understand why neural network can generalize well with only finite training samples is still an open problem.
> >
> >     For the experiments, we will run more experiments on smaller training sets and report the results in the revision.
> >
> > - W4: Justification / intuition for Theorems is limited in the main paper.
> >
> >     We will clarify this in the revision. For the main theorem (Theorem 3), it shows that if the SSL objective is optimized, then no matter what model was used to generate the data, (low degree polynomial of) the true posterior of this model is going to be a linear function of the representation learned by the SSL algorithm. As we explained in the general response, this shows that SSL learns a good representation that contains information about the posterior for the correct model that was used to generate the data. Intuitively, the reason this theorem is true is that the best way to predict the next word (or next $t$ words) is to first compute the posterior of the topic proportions and use this topic proportion to generate the new words.
> >
> >     For the tightness of Theorem 4, we do not expect the bound given by Theorem 4 is tight in practice as the constant in this bound are likely to be loose. However, we believe that the approximation error is $O(\epsilon)$ is informative, since it shows the robustness of SSL representation.
> >
> > - Detailed Feedback on Presentation.
> >
> >     Thanks for the detailed feedback. We will fix them accordingly in the revision.

---

> > > ### Comment · Reviewer_KozZ · 2021-11-24
> > > **Detailed response to the previous issues I raised**
> > >
> > > I'll respond to each of points made by the authors (W1-W4) here. (I'll include a revised "higher-level" review with my remaining concerns elsewhere).
> > >
> > > ### W1: Procedure for reconstructing the topic posterior is unclear
> > >
> > > OK, thanks I understand now that the posterior distribution itself is never estimated (that's not the goal), it is only summary statistics of this distribution (like the mean or variance) that can be estimated.
> > >
> > > ### W2: Steps difficult to understand / replicate
> > >
> > > Thanks for these responses, my understanding has improved. I still think the latest draft has some issues that will cause future readers trouble (and are cause for concern in accepting the present form):
> > >
> > > 1) The current versions "Neural network models" paragraph is still confusing, seems like every NN takes an ordered list as input. Please revise
> > > 2) For any dataset included, please do report at least the #documents, #vocab terms, and typical document length in the main paper. Leaving these to the supplement prevents readers appreciating the big picture of your experiments.
> > >
> > > I also think that the description of "TV" distance should be changed, so that instead of describing it as "Total Variation (TV) distance between the recovered topic posterior and the ground truth topic posterior", it is instead described as the distance between the *posterior mean vectors* (recovered vs true).  The current text makes it seem like the procedure here produces a distribution, but in fact it only produces the mean of the vector w.
> > >
> > > ### W3 Unclear what theorems / methods can say about finite training sets
> > >
> > > I'm glad the authors ran more experiments, but I think that the 120k training document setting is quite large for many text applications.
> > > Note that typical Bayesian topic models (like LDA) work pretty well even when there are only a few thousand documents, and even when some documents are quite short.
> > >
> > > I guess my worry here is that the settings in which these theorems work well are really the "large data limits", where each document is quite large and we have many many such documents, and thus ML point estimates are expected to be quite good. This is fine, but I think should be stated a bit more clearly than is currently.
> > >
> > >
> > > ### W4 Justification / intuition for Theorems
> > >
> > > Thanks, glad to see some thoughts here. I think this issue is resolved, it doesn't impact my concerns about the paper much anymore.

---

> > > > ### Author Response · Authors · 2021-11-27
> > > > **Response**
> > > >
> > > > We appreciate reviewer's effort to provide a detailed response. We are glad to see some of the concerns have been addressed (W1 and W4). We try to address the remained concerns as below.
> > > >
> > > > - W2.
> > > >
> > > > Thanks for the suggestions. We will try to change them accordingly in the revision.
> > > >
> > > > - W3.
> > > >
> > > > As we have clarified in the general response, we do not claim SSL can achive state-of-the-art performance (both in terms of accuracy and training sample size). Instead, our goal is to show SSL can extract useful information about the topic posterior without knowing the correct prior. While intuitively ML point estimate  is accurate in the ''large data limit'', we would like to point out that the fact that this particular ML estimate (produced by SSL) can extract useful information about topic posterior is actually surprising (which is the main focus of this paper).

---

> ### Comment · Reviewer_KozZ · 2021-11-24
> **Revised Opinion and List of Concerns following initial author response**
>
> After reading the other reviews and the author response, I have revised some opinions, which I'll summarize here. Bottom line: I think this is promising work, but I'm unfortunately not convinced the present paper is suitable for publication at present.
>
> I'll agree with the authors that the main contribution of the paper is trying to explain how self-supervised methods can extract meaningful representations regardless of the underlying generative model of bag-of-words text data. I don't mind if results are not "state-of-the-art" (the goal here is not to beat BERT), though I do expect experiments to be *fair* (put methods on an equal footing when possible) and validate the claims of the paper.
>
> The major strengths I see in the paper continue to be its interesting topic, coverage of a broad set of topic models and SSL objectives (potentially broad applicability), and its theoretical results (showing the surprising ability for a "predict-one-heldout-word" representation to be useful for estimating the mean of a posterior for any topic model).
>
> However, the major weaknesses in my revised view are the real-data experimental justification for the paper's claims as well as consistent presentation quality issues that hinder a new reader's understanding and reproducibility. I still also have concerns about how this theory applies to smaller real datasets (see W3 from my original review), but this is less important to me than the experimental and presentation issues.
>
> ## Issue 1: Real Data Experiments (Sec 6) don't seem connected to the paper's main claims, lack baselines to be connected to those claims, and don't give some baselines a fair shot
>
> In Sec 6, the authors show how using a representation f(x) trained with SSL can be used for downstream semi-supervised classification task that assigns documents into one of 4 topics. I understand that Sec. 6 seems intended to show that the proposed representation can succeed on a real downstream task. That's fine, but to me, this isn't too surprising (of course a large network trained to do something useful can be repurposed for another similar task).
>
> I agree with other review that "This [Sec 6] feels like a bit of a non-sequitur given the rest of the paper"... If we look at the current abstract, no claim there is really supported by this section (there's no topic modeling here, just classification using different representations). These experiments doesn't help with any of the study's main claims about being "better than" misspecified topic models. I think either the abstract should be revised to adjust claims (to be much broader than topic models), or this section should be revised to include topic model baselines.
>
> I further agree with the reviewer HibN above that this comparison would be strengthened by comparing to one or more recent supervised or semi-supervised topic models (see the list above, there is code available for each of these methods).  This would help us understand the abstract's current claim that the presented SSL methods would *outperform* "posterior inference using mis-specified model". Without this comparison, there's no "real data" evidence for the claims in the abstract.
>
> Furthermore, the comparisons in Sec. 6 seems *unfair* to some baselines. The proposed reconstruction objective is given a neural net with size 4096, while the word2vec baseline has embedding dimension of only 300. Best practice would be to set these sizes equivalent, or (better yet) allow word2vec to be much larger (since it is a single layer not many layers like the present method). (Perhaps 300 was selected in a hyperparameter search, but the paper does not say so).
>
>  ## Issue 2: Presentation issues inhibit understanding / reproducibility
>
> I appreciated the authors' willingness to respond to my original detailed questions about method details (W1-W2) and revise the paper accordingly. I do hope they continue to do so.
>
> Unfortunately, the current paper version still is tough for a reader to understand, to a degree that I think another round of revision with reviewer feedback would be useful.
>
> Examples:
>
> - the main paper needs a concrete description of how the posterior mean vector is estimated for the experiments in Sec 5: $E[w|x] = A^{\dagger} f(x)$. The text discussion of this method below one of the theorems is nice and general, but adds a lot of mathy-ness that makes it hard to understand the practical implications.
> - Sec 6 really obscures the fact that what is being evaluated is a *semi-supervised* classification task, and that some methods have a very different representation size than others. These are basic details that belong in the main paper, not an appendix. See my detailed responses below for additional examples.

---

> > ### Author Response · Authors · 2021-11-27
> > **Response**
> >
> > We appreciate reviewer's effort to provide a detailed response. We are glad to see some of the concerns have been resolved. We try to address the remained concerns as below.
> >
> > - Issue 1.
> >
> > As we have said at the beginning of Section 6 in the revised paper, the goal of Section 6 is to study whether SSL can learn some reasonable representation beyond topic models, when comparing with some baselines. Thus, there are no correct and mis-specified models as there are no topic models at all. We do not claim SSL can achieve state-of-the-art performance and outperform the recent methods, which is not the goal of this section and the current paper.
> >
> > For the embedding dimension of word2vec, we would like to note that even for the current small embedding dimension, its performance is not good when the supervised training sample size is small. Intuitively, the performance would become worse when the dimension becomes larger, since the supervised training sample size is too small (smaller than the embedding dimension).
> >
> > - Issue 2.
> >
> > Thanks for the suggestions on the presentation of current paper. We will try to change them accordingly. The practical implication of Theorem 3 is mostly that minimizing this objective will give useful representations (if the data can at least be approximated by topic models). We will clarify that in later versions.

---

> > > ### Comment · Reviewer_KozZ · 2021-11-29
> > > **Further thoughts on Issue 1**
> > >
> > > ### RE Issue 1 and word2vec's small size
> > >
> > > I guess I disagree that it is a foregone conclusion that "Intuitively, the performance would become worse when the dimension becomes larger". If you have good hyperparameter search on your prediction model, you should be able to mitigate severe overfitting. (Only an experiment can tell us if it really will improve or get worse slightly or get worse by a lot).
> > >
> > > Plus, Fig 2 allows up to 4000 training examples, so seems like the current choice of 300 could be restrictive.
> > >
> > > I really think you should redo this experiment with much more fair treatment and attention on baselines.
> > >
> > >
> > > ### RE Issue 1 and why I think experiments need to assess topic models### RE Issue 1 and why I think experiments need to assess topic models
> > >
> > > Here's my thinking about why the real-world experiments need to compare to *some* topic models.
> > >
> > > First, this paper is sub-titled "SELF-SUPERVISED LEARNING FOR TOPIC MODELS". So a reader should expect to find a real-world experiment about topic models.
> > >
> > > Second, the abstract claims that:
> > >
> > > > Empirically, we show that the same objectives can perform competitively against posterior inference using the correct
> > > model, while outperforming posterior inference using mis-specified model
> > >
> > > However, this claim -- that the proposed objectives can outperform posterior inference in a misspecified (topic) model -- is never validated on real-world data.
> > >
> > > I guess I see two ways forward:
> > >
> > > - rewrite the title and abstract, and make the paper much more broad in scope (about SSL for bag-of-words text data)
> > > -  cut out the current Sec 6 entirely and expand on the topic model posterior assessments in Sec 5
> > >
> > > Both involve significant changes that I think would need another round of review.

---

> > > > ### Author Response · Authors · 2021-11-29
> > > > **Response**
> > > >
> > > > - word2vec size
> > > >
> > > > We would like to note that in the supervised phase, we only do a linear classifier on the representation provided by these methods. If the dimension is high and number of training samples is small, then this is likely that the performance is not good even with a good hyperparameter search since sample size is smaller than dimension. Therefore, simply increasing the dimension of representation would be very likely to worse the result as it makes the ratio between sample size and dimension becomes even more small. We agree that the performance might be increased when sample size is 4000, but the result would become worse for the case of 300 sample size. It seems hard to balance these two cases so we currently choose to use a small dimension. We will also check this empirically.
> > > >
> > > > - real world experiments
> > > >
> > > > We respect reviewer's opinion and would like to thank reviewer for the suggestions and thoughts on the presentation. However, as we clarified in the previous post, the goal of this section is to study SSL beyond topic model, so there is no ''correct'' topic model in this setting. On the other hand, we are willing to adjust the abstract to add ''on synthetic data'' to the sentence that the reviewer refering to.

---

> > > > > ### Comment · Reviewer_KozZ · 2021-11-30
> > > > > **Thanks for your participation and good luck with the project**
> > > > >
> > > > > Just wanted to say thanks to the authors, I learned more than usual from this discussion period and I am glad for your participation.
> > > > >
> > > > > I wish you the best for this project...  I know getting constructive criticism is frustrating sometimes but I do think your paper will be stronger and more widely appreciated if you make the baseline comparisons more extensive and fair and improve the reproducibility further.
> > > > >
> > > > > RE baselines: I am glad you'll look into the word2vec size empirically.
> > > > >
> > > > > RE abstract: Thanks, I would like to see the abstract specify that the topic model claims are all on synthetic data.

---

### Official Review · Reviewer_HibN · 2021-11-05

**Correctness:** 3
**Technical Novelty And Significance:** 3
**Empirical Novelty And Significance:** 3
**Recommendation:** 6
**Confidence:** 3

**Main Review:**

Strengths:

The main results here seem novel and potentially useful. The proposed method is a new way to perform efficient approximate inference for topic models with some guarantees.

Compared to previous work the authors provide a new self-supervised learning technique for topic models with its own theory.

The synthetic experiments show that the approach is useful when a misspecified model is used.

The real data experiments show that this self supervised learning technique is useful for feature extraction.

Weakness:

I feel that some of the results of this work could use further explanation and that some of the experiments are confusing or underwhelming:

I do not feel I was able to devote adequate time to understanding the details of the results shown in section 3 and I’d like to continue looking in to it. For the third paragraph of section 3, could the authors give additional intuition for how this approach gives the correct expectation of the posterior seemingly without knowledge of the prior p(w)?

Figure 1 provides very little information, as it seems to only reflect a single synthetic document. I’m not sure what I’m supposed to take away from it.

Figure 2 seems like a poor way to present this information. Even simple table would probably be more interpretable and precise.

Under: Robustness of self-supervised learning. What is the “traditional topic inference” used? Simply the sampling approach discussed above? It is also strange that the results are mostly shown comparing across misspecified models. I would expect to see comparisons between other types of inference (MAP, variational) using correctly specified models. Comparing explicitly misspecified models for synthetic data feels a bit contrived.

Why are the results on real data only about semi-supervised learning? This feels like a bit of a non-sequitur given the rest of the paper. I would expect to see at least qualitative results showing inference with this method on real data.

There has been quite a bit of prior work on supervised and semi-supervised topic models. I would expect at least some of these methods to be represented in section 6. Some examples being: https://proceedings.neurips.cc/paper/2007/file/d56b9fc4b0f1be8871f5e1c40c0067e7-Paper.pdf, https://www.jmlr.org/papers/volume13/zhu12a/zhu12a.pdf, http://proceedings.mlr.press/v84/hughes18a.html

Generally the focus of this paper is on inference. Can this approach be applied within a learning algorithm for topic models?

**Summary Of The Paper:**

This paper introduces a new method for performing inference on topic models, based on self supervised learning. Their approach is generalizable to multiple types of topic models. Specifically, the authors show that the expectation of some polynomial function of the topic posterior will be a linear function of the output of function that optimizes the self-supervised learning objective. The authors explore 2 types of self-supervised learning objectives, one based on reconstruction and one based on contrastive learning that was explored in prior work. The authors provide proofs for their main result as well a result robust to approximate minimization of the self-supervised objective. In their experiments the authors show that their approach outperforms inference with a misspecified model. The authors also run semi-supervised learning experiments on their approach.


**Summary Of The Review:**

Overall the results in this work are interesting and possibly useful, but I feel that there are some lingering questions about applicability, particularly as the experimental results are underwhelming.

---

> ### Author Response · Authors · 2021-11-16
> **Response**
>
>
> We would like to thank reviewer for providing the detailed response. We will try to address the concerns below. We also make several clarifications in a separate post to address some common concerns for all the reviewers.
>
> - More intuition on Theorem 3 (the third paragraph of section 3).
>
>     The theorem shows that if the self-supervised learning (SSL) objective is optimized, then no matter what model was used to generate the data, (low degree polynomial of) the true posterior of this model is going to be a linear function of the representation learned by the SSL algorithm. As we explained in the general response, this shows that SSL learns a good representation that contains information about the posterior for the correct model that was used to generate the data. Intuitively, the reason this theorem is true is that the best way to predict the next word (or next $t$ words) is to first compute the posterior of the topic proportions and use this topic proportion to generate the new words.
>
> - Little information in Figure 1.
>
>     The purpose of Figure 1 is to give an illustration of the CTM and PAM setting that we consider in the experiments. As we described in the second paragraph in Section 5.1, we construct a setting where within the group of 4 topics 0,1,2,3, topic pairs (0,2) and (1,3) are highly correlated (as specified by the prior), while topics 0 and 1 share many words. We hope Figure 1 can help readers to understand our setting more easily.
>
> - Presentation of Figure 2.
>
>     Thanks for the suggestion. We will change Figure 2 to a table to present our results.
>
> - ``Under Robustness of self-supervised learning ...''
>
>     By ``Traditional topic inference'' we meat to say the topic inference method that tries to compute the topic posterior when a prior is given, such as sampling approach. As we clarified in the post for all reviewers, the focus of current paper is not to achieve a state-of-the-art performance in the topic inference problem. Instead, our aim is to show  SSL is more robust to model misspecification than traditional topic inference method like sampling approach. This is the reason why we compare the results with misspecified models.
>
> - Semi-supervised learning in real data experiments.
>
>     As we clarified in the post for all reviewers, the problem that we are interested in is not inference problem, but to understand why SSL can provide useful information for the downstream task. In the real data experiments, our goal is to show the representation given by SSL can help to solve the downstream task and achieve a reasonable performance.
>
> - Compare with other supervised and semi-supervised topic models.
>
>     As we explained in the general response, the problem that we studied in current paper is not the inference problem, and we do not claim that SSL can perform better than other methods. Our focus is to understand why SSL can provide useful information for the downstream task and show that even without a correct prior, SSL can still achieve a reasonable performance.
>
> - Using current approach within a learning algorithm for topic models.
>
>     Again, as we have clarified, the main contribution of current paper is not to give a new method to do inference, but to understand why SSL can extract useful information for the downstream task.

---

> > ### Comment · Reviewer_HibN · 2021-11-27
> > **Response**
> >
> > Thank you to the authors for their detailed responses!
> >
> > In general, the changes made that make the goal of the paper more clear are very useful. Reading this work as an analysis of SSL rather than as primarily an inference technique for topic models does make many of the authors choices in presentation more reasonable. I'm raising my score to reflect my new opinion.
> >
> > * Re: More intuition on Theorem 3
> >
> > I think this does help to clarify the purpose somewhat. I feel like I'm still missing something obvious as to how $E[w|x]$ can be found without knowledge of the prior $p(w)$, as claimed in that section. In the supplement under lemma A.2 you say: "It is easy to see that the word posterior distribution is $AE_w[w|x]$". Could you explain this fully?
> >
> > **Edit:** I guess the obvious thing I'm missing is that $p^*(x)$, as it's called in the supplement, is not actually the posterior predictive distribution for a given topic model & document, it's something else. Am I right in saying then that the $E[w|x]$ is not the mean of the posterior for a particular topic model? I don't really have time to go through all the math, but this is confusing me. Could the authors elaborate on what the paragraph at the end of page 4 is actually saying?
> >
> > * Re: ``Under Robustness of self-supervised learning
> >
> > I understand that the purpose of the work is not improved inference, so I agree that comparisons to other inference techniques isn't strictly needed. That said, for the toy data section you are still evaluating it as essentially an inference technique, so it seems reasonable to include a comparison to widely used approximate inference techniques, such as VB.
> >
> > * Re: Little information in Figure 1.
> >
> > This does clarify the purpose, I think it is a reasonable figure to include.
> >
> > * Re: Presentation of Figure 2.
> >
> > Thank you for making this change. I do think table 2 is clearer than the previous figure.
> >
> > * Re: Compare with other supervised and semi-supervised topic models.
> >
> > With my better understanding of the goals, perhaps this is less necessary, but I still think it would be illuminating. This paper looks at SSL through the lens of topic models, and specifically discusses that topic models can be used as a feature extractor, so why not compare to traditional topic model inference for this use case? Comparing to semi-supervised topic models could make this point even more strongly as they combine the learning of topic models as a feature extractor while being informed by the supervised learning task.
> >
> > * Re: Using current approach within a learning algorithm for topic models
> >
> > Agreed that this is likely out of scope for the goals of this paper, I'm more curious if this would work.

---

> > > ### Author Response · Authors · 2021-11-29
> > > **Response**
> > >
> > > We appreciate reviewer's effort to provide detailed response. We are glad to see some of the concerns have been addressed and the reviewer is willing to raise the score. We will try to address the remained concerns below.
> > >
> > >
> > > - More intuition on Theorem 3
> > >
> > > We will try to explain it in the case of $t=1$. In this case, $p^*(x)$ is a $V$-dimensional probability vector and its meaning is word posterior distribution (i.e., the $i$-th entry of $p^*(x)$ is the probability of word $i$ will show up given the document $x$). $E[w|x]\in \mathbb{R}^K$ is the posterior mean of topic posterior. Recall that in topic models, given the topic proportion $w$ of the document, one can view the word is sampled from the distribution $Aw$. Similarly, since $p^*(x)$ is the word posterior distribution and $E[w|x]$ is the posterior mean of topic posterior, we have that $p^*(x)=AE[w|x]$.
> > >
> > > - Under Robustness of self-supervised learning
> > >
> > > In the current version of paper, we added a comparison of SSL with Variational Inference on the recovering major topics to Appendix C.3 (Table C.3) to show the performance of SSL is competitive with variational inference with correct prior.
> > >
> > > - Compare with other supervised and semi-supervised topic models.
> > >
> > > Thanks for the suggestion. We choose not to compare with other methods in the current paper so that the main contribution of this paper (to understand why SSL can extract useful information for the downstream task) can be emphasized and we do not want to claim SSL is better than all other methods.
> > >
> > > - Using current approach within a learning algorithm for topic models
> > >
> > > Unfortunately we haven't tried this direction and do not know the answer. We would like to view this as an interesting future direction.

---

> > > > ### Comment · Reviewer_HibN · 2021-11-29
> > > > **Further response**
> > > >
> > > > Thank you again for the response!
> > > >
> > > > I think the sticking point for me with regards to section 3 is that the implication seems to be that this procedure holds for finding $E[w|x]$ for any particular ("general") topic model, when my understanding is that this is not true. I haven't worked it out, but I'm guessing that it finds $E[w|x]$ specifically for the "pure" topic model or a topic model with an uninformative prior (i.e. one where $p(w)$ is uniform on the simplex). Am I wrong about this? If not, what is the implication for theorem 3, does that still hold for any topic model?
> > > >
> > > > **Edit:** To further this point, why isn't the "pure" topic model shown as a TV distance method in any of the tables?

---

> > > > > ### Author Response · Authors · 2021-11-29
> > > > > **Response**
> > > > >
> > > > > Thanks for the question.
> > > > >
> > > > > The results in section 3 hold for any general topic model. The reason is that (1) we can have $p^*(x)$ for any general topic model as long as $p^*(x)$ is the optima of the objective; (2) $p^*(x)=AE[w|x]$ holds for any general topic model. Let us know if this helps.
> > > > >
> > > > > For 'Edit' question:
> > > > > If the question is about Table 2, as we have explained in the paragraph 'Robustness of self-supervised learning', pure-topic prior gives invalid results for documents with mixed topics. Therefore, we do not report the result.

---

> > > > > > ### Comment · Reviewer_HibN · 2021-11-29
> > > > > > **Response**
> > > > > >
> > > > > > I hate to argue, but doesn't $p(y|X)$ (probability of the new word) depend on the prior for the topic distribution? So how can a function that depends only on the document, $f(x)$, possibly give the right answer in all cases?
> > > > > >
> > > > > > As to the edit, I suppose that does make sense, I didn't think about it.

---

> > > > > > > ### Author Response · Authors · 2021-11-29
> > > > > > > **Response**
> > > > > > >
> > > > > > > $f(x)$ indeed depends on the prior of topic proportion and the reason is that $f(x)$ is the optima of the objective. For different prior, the data x,y are different, so the optima of the objective is going to be different (as the objective is a expectation over the data distribution). In this way, $f(x)$ implicitly depends on the prior information and it is a function of $x$ when we focus on a fixed topic model.
> > > > > > >
> > > > > > > Hope this could clarify the confusion.

---

> > > > > > > > ### Comment · Reviewer_HibN · 2021-11-29
> > > > > > > > **Response**
> > > > > > > >
> > > > > > > > Oh, I finally see the what I'm missing. I was missing that the data used to train $f(x)$ actually has to be generated from the true topic model that it's being used for inference for, so the $E_{x,y}[\cdot]$ in the reconstruction objective is over data drawn from the topic model, not over the data used to train the topic model.
> > > > > > > >
> > > > > > > > This is obvious in hindsight, but I definitely feel it could be better explained in the paper. With this finally making sense to me, I'll retain my raised score.

---

### Author Response · Authors · 2021-11-16
**General Response**


We thank all the reviewers for their time to provide detailed reviews. We would like to clarify several points below to all reviewers. Detailed response to each review will be posted separately after the corresponding review.

- Main contribution of the paper. We would like to emphasize that our goal is not to design a new state-of-the-art algorithm nor to show self-supervised learning (SSL) can achieve the state-of-the-art performance for topic inference problem. Instead, our main contribution is to show that SSL can be oblivious to the specific topic model, and hence is less susceptible to model miss-specification. In other words, we want to show that SSL can extract useful information for a general topic model no matter what underlying topic model is used to generate documents. We will make it more clear in the revision.


- For the same data, different topic models/priors would make different predictions. Since our main goal is to show that self-supervised learning is oblivious to the way that the data is generated, our approach is to show that SSL can correctly predict the posterior distribution in a semi-supervised setting no matter what topic model is used to generate the data. More precisely, our SSL objectives (both the reconstruction based objective and the contrastive objective) will learn a representation of the documents that is independent of the topic model that is used to generate the data. One can view this step as an unsupervised pretraining. If the loss of the SSL objective is low, no matter what topic model was used, its posterior information can be extracted by using a linear function on the representation. This linear function can be learned using small amount of labeled data (hence the whole task is a semi-supervised learning problem). We will modify intro/preliminary sections to make this more clear.

---

### Author Response · Authors · 2021-11-23
**Paper Updates**

We have updated our paper to address the concerns from reviewers. Specifically, we made following changes:

- Edited the introduction section to highlight the question we are solving and our contributions. (all reviewers)

- Added a paragraph of "semi-supervised learning setup" in preliminaries to highlight the setup that we are considering. (all reviewers)

- Edited explanation after Theorem 3 to give more intuitions. (reviewer HibN, KozZ)

- Explained why we are doing the real data experiments at the beginning of Section 6. (reviewer HibN)

- Added all results with 95\% confidence interval in Section 5 and Appendix C.3 to show the stability of our results on the test set. (reviewer KozZ)

- Added the performance of SSL with 120k ("finite") training documents to Appendix C.3 (Table C.1 and C.2) to show SSL has a reasonable performance on ''finite'' training set. (reviewer KozZ)

- Added a comparison of SSL with Variational Inference on the recovering major topics to Appendix C.3 (Table C.3) to show the performance of SSL is competitive with variational inference with correct prior. (reviewer HibN)

- Fixed several typos pointed out by the reviewers. (all reviewers)

Please let us know if there are any further concerns/comments.

---

### Decision · Program_Chairs · 2022-01-20

**Decision:**

Reject

**Comment:**

This paper generated a large amount of discussion.  Three reviewers were marginally above and one marginally below.  The paper presents an intriguing relationship between self-supervised learning and topic model inference that extends earlier work of Tosh.  The result seems to be subtle because there was considerable discussion with the authors wrapping up with a reminder of what the main goal is:  SSL can achieve the state-of-the-art performance for topic inference problem, moreover (main goal) SSL can be oblivious to the specific topic model.  This is indeed intriguing.  But with all the discussion, and one persistent negative reviewer, I feel the paper needs to be polished.  Given the theorem gives a testable statement, I don't see why experimental results cannot be done for 4 different real data sets, to give us more confidence.